# Exploring Cellular Protein Localization Through Semantic Image Synthesis

## Abstract

Cell-cell interactions have an integral role in tumorigenesis as they are critical in governing immune responses. As such, investigating specific cell-cell interactions has the potential to not only expand upon the understanding of tumorigenesis, but also guide clinical management of patient responses to cancer immunotherapies. A recent imaging technique for exploring cell-cell interactions, multiplexed ion beam imaging by time-of-flight (MIBI-TOF), allows for cells to be quantified in 36 different protein markers at sub-cellular resolutions *in situ* as high resolution multiplexed images. To explore the MIBI images, we propose a GAN for multiplexed data with protein specific attention. By conditioning image generation on cell types, sizes, and neighborhoods through semantic segmentation maps, we are able to observe how these factors affect cell-cell interactions simultaneously in different protein channels. Furthermore, we design a set of metrics and offer the first insights towards cell spatial orientations, cell protein expressions, and cell neighborhoods. Our model, cell-cell interaction GAN (CCIGAN), outperforms or matches existing image synthesis methods on all conventional measures and significantly outperforms on biologically motivated metrics. To our knowledge, we are the first to systematically model multiple cellular protein behaviors and interactions under simulated conditions through image synthesis.

## 1 Introduction

### 1.1 Biological Role of Cellular Proteins

Cell-cell interactions within the tumor microenvironment have been implicated in many facets of cancer pathogenesis and treatment. Most prominently, tumor cell evasion of immune surveillance (Jiang et al., 2019), tumor metastasis (Nishida-Aoki & Gujral, 2019), and efficacy of cancer immunotherapies (Lau et al., 2017) have all been closely linked to the relationships between immune and cancer cells. These types of cell-cell relationships are generally governed by the interactions of cell surface proteins which drive cell behavior, gene expression, and survival. One of the most prominent examples of cellular proteins influencing disease progression is the case of PD-1/PD-L1. PD-L1 is a protein often overexpressed on tumor cells and has the capacity to bind PD-1 on local T cells to downregulate their anti-tumor immune responses (Iwai et al., 2002).

Antibodies which interrupt the interaction of PD-L1 and PD-1 and allow the immune system to attack tumor cells, have become clinically influential treatments for a variety of cancers (Pardoll, 2012). This example highlights the value of accurately predicting cellular protein patterns which play key roles in disease processes. Exploring protein localizations in a multi-cellular system represents a challenge for which deep learning models are uniquely suited. However, to our knowledge no image-based deep generative models have utilized semantic image synthesis to produce accurate predictions of these biological phenomena.

### 1.2 Framework and Novelty of a Predictive Model

A meaningful exploration of cell-cell interactions, particularly in the tumor microenvironment, requires a thorough understanding of the proteins expressed on and within a cell and its neighborhood. Multiplexed ion beam imaging by time-of-flight (MIBI-TOF) represents a novel technology that can accurately quantify and spatially resolve cellular protein expressions at the single cell level within

tissue samples. Given a tissue sample that is first stained with protein-specific antibodies tethered to elemental metals, MIBI-TOF bombards the sample with atomic ions (i.e. $O_2^+$) from a primary ion beam. This causes the release of elemental isotopes and tissue-specific material which can be quantified in a mass spectrometer (Angelo et al., 2014). The cellular proteins characterized by this technique indicate specific cell types (i.e. immune cells, tumor cells), cell status (i.e. markers of proliferation), and immunomodulation. Figure 1 (A) shows an example spatial orientation of cell types and some selected cellular protein expressions. Here, we propose a novel protein based attention

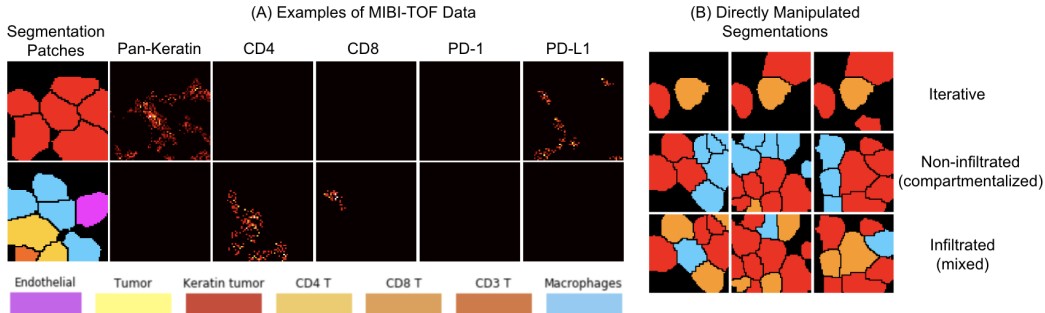

Figure 1: (A) Example segmentation patches (left column) and protein expressions (other columns). (B) Examples of how segmentation maps can be synthetically altered to pose different counterfactual biological scenarios for *in silico* hypothesis testing.

mechanism for a convolutional Generative Adversarial Network (GAN) with the capacity to provide accurate conditioned predictions of cellular protein localizations. The model, Cell-Cell Interaction GAN (CCIGAN), learns a many to many mapping between different cell types and different protein markers. CCIGAN is trained on semantic segmentation maps of cell tissue samples to identify and associate cell type, shape, and the identity of its cellular neighbors with cellular protein expressions for each cell. The network was then applied to simulated segmentation maps to understand how cells vary their protein localizations.

CCIGAN demonstrates its ability to probe elements of oncologic processes by:

1. Independently recapitulating established patterns of biological phenomena, thereby demonstrating the accuracy of its predictive power.

2. Yielding quantitative and spatial information on specific cellular protein modulations as a result of immune cell - cancer cell interactions.

3. Allowing for biological data to be generalized beyond traditional *in vitro* scenarios where specific, rare, or hypothetical cell-cell events can be posed in counterfactual scenarios and the results of their interactions predicted.

The ability for CCIGAN to generate subcellular protein predictions represents a step forward in the ability to understand cellular relationships within a microenvironment. Rather than assessing *in vitro* incidence of cell interaction phenomena as is done by MIBI-TOF, CCIGAN allows for hypothetical biological situations to be generated. Figure 1 (B) illustrates examples how a segmentation map can be directly manipulated to pose various biological scenarios. In other words, individual cellular responses to challenge or proximity of cells of any identity can be assessed without having to seek this specific occurrence within the available biological tissue sample. These capabilities allow the model to provide insight on hypothetical interaction events which shed insight on disease pathogenesis and which cannot be feasibly achieved by biological investigation alone.

MIBI-TOF is a labor intensive process which can make the data collection process cumbersome. Deep learning models such as CCIGAN have the potential to be useful tools which can extend on MIBI-TOF capabilities and result in more rapidly synthesized data. CCIGAN, once trained on MIBI-TOF data sets and properly replicating known cell-cell interaction patterns, can be used as a robust means to provide high throughput readouts on myriad single cell scenarios which may normally take many iterations of MIBI-TOF experiments to investigate. Recent work by (Wu et al., 2019) demonstrated the utility of conditional GANs for adding a depth dimension to images; we

demonstrate learning contextual cell-cell interactions, facilitating rapid hypothesis testing to assess biological environments. This provides valuable insight into a complex system that normally would have taken resource-intensive wet lab experiments to interrogate.

## 2 RELATED WORK

We are interested in the task of generating biologically consistent expression patterns of cellular proteins given a segmentation map of cell neighborhoods. Specifically, we want to learn a generative model that simultaneously produces high quality maps of protein expression for individual cells that are probabilistically consistent when conditioned on the same factors, e.g. similar cell neighborhoods should produce similar expression patterns. Such generative models typically take the framework of a generative adversarial network (GAN) (Goodfellow et al., 2014; Brock et al., 2018) or a variational autoencoder (VAE) (Kingma & Welling, 2013).

Within these generative modeling techniques, image translation focuses on learning a many to many mapping to transform data from one domain to another. One approach is to model the translation task, known as image synthesis, with conditioning such as Pix2Pix (Isola et al., 2016), Pix2PixHD (Wang et al., 2018a) and CycleGAN (Zhu et al., 2017). By using image synthesis, a model is able to distill more information from a segmentation map to various protein channels. As MIBI-TOF measures information such as protein localizations at a subcellular resolution, framing the problem through an image synthesis framework allows a model to see how predictive neighborhoods and cell types are of a cell's phenotype.

Recently, Park et al. (2019) proposed spatial adaptive normalization (SPADE) to synthesize images using segmentation maps by learning fully convolutional normalization parameters based on their segmentation conditioning. For each layer, semantic information is retained by allowing the network to learn from the segmentation map directly and modulate the current layer. They demonstrate impressive quality and diversity in generated images, especially modeling objects in context. Using SPADE normalization, CCIGAN is able to condition on surrounding cell neighbors and capture cell-cell interactions in a local receptive field. Despite image generation, there has been no work done towards understanding such generations in context, particularly in biological images.

Attention layers (Vaswani et al., 2017; Wang et al., 2018b; Zhang et al., 2018) have also been added to generative models with great success. Self-attention was initially proposed in machine translation tasks to help model long distance dependencies that occur frequently in language (Sutskever et al., 2014), which led to the idea of external memories as a persistent state to model long range dependencies (Sukhbaatar et al., 2015). While convolutional neural networks are apt at exploiting local structures of patches, they may struggle to model global structures. For this reason visual attention allows the network to enhance activations in interesting parts of an image. Goodfellow et al. (2014) and Brock et al. (2018) both achieve state-of-the-art unconditional generative modeling using self attention GAN (Zhang et al., 2018). We propose a specialized attention module conditioned on different proteins to mimic real world protein markers.

## 3 DATASET

### 3.1 MULTIPLEXED ION BEAM IMAGES (MIBI)

All experiments are performed on data obtained through MIBI-TOF characterized tissue samples collected from triple-negative breast cancer (TNBC) patients. By simultaneously imaging over 36 protein markers, MIBI-TOF is able to identify cell type as well as provide detailed information of sub-cellular structure, cell neighbors, and interactions in the tumor microenvironment across these different marker settings. Each of these markers $m \in \{1, ..., M\}$, given as a channel taking on real values continuous in $[0, 1]$ at each $(x, y)$ coordinate, demarcates a different cellular subtype and furthermore, is indicative of the functional properties of a cell. While MIBI-TOF is capable of 36 different markers, we chose markers exhibiting cellular protein localization and disregarded markers that were blank or used simply for cell-typing [1] resulting in $M = 24$. A full list of these markers is given in A.8.

---

[1] Some channels were completely blank and others were indicator channels, i.e. CD45, a binary indicator for an immune cell.

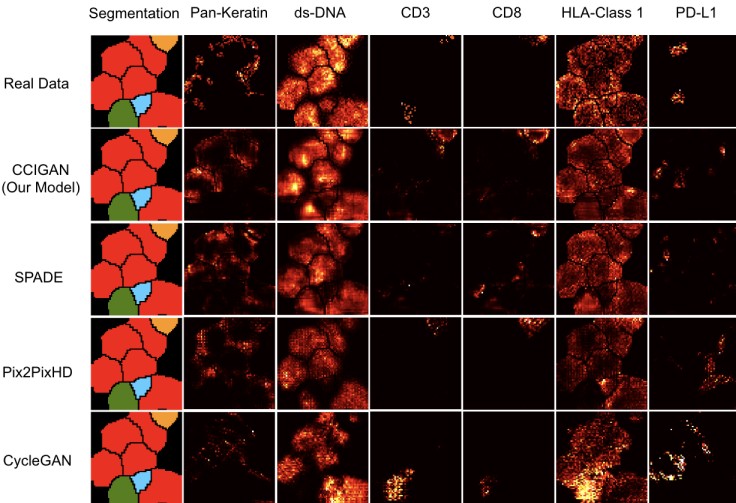

Figure 2: Examples generated from a segmentation for certain channels for different models. The segmentation patch is the one hot encoded patch collapsed and colored into 1 channel. The horizontal labels represent protein markers and the vertical labels are each of the generative models.

The MIBI-TOF data collected from the TNBC tissue allows for the differentiation between a wide variety of cell types within the samples. For instance, cells positive for the marker CD3 could be identified as T cells, and then subdivided into cytotoxic or helper lineages by the presence of markers CD8 or CD4, respectively. Tumor cells could be identified by markers such as pan-keratin and overexpressed beta-catenin. Along these lines, a wide variety of cellular proteins identified by MIBI-TOF could characterize cell interactions as well as immunomodulatory processes occurring within the microenvironment.

MIBI-TOF data is fundamentally different than typical RGB images. This poses unique challenges in image modeling and characterization. Such challenges stem from each marker being conditionally expressed on the cell type in its respective channel $m$. Using a simplified 3 channel multiplex setting, a T cell expresses signals in the CD3, CD8 channels (indicators for immune cells) but not in a pan-keratin channel (indicator for tumor cells). Another problem is the sparsity of the data, meaning either some expressions for rare cell types are rarely observed or have weak signals. For example, Figure 2 includes a segmentation of a CD8 T cell (orange, top right corner), where other models fail in generating correct CD3 expression, if at all[2]. Lastly, the noisy nature of the data leads to inaccurate cell type classifications, creating inconsistent pairs of labels and outputs during training.

These issues make it especially difficult for an RGB multihead decoder to output multiple channels in a biologically accurate manner. Without addressing multiplexed data, a decoder would equally attend to every location of the current latent representation, even if it is irrelevant to the current protein. Furthermore each protein channel in the output has its own sensitivities to signal intensities and noise, suggesting each channel requires a unique prior and that equal attention would be problematic. It follows that special care must be given towards modeling specific channels and the multiplexed nature of MIBI-TOF images.

## 3.2 DATA PROCESSING

MIBI-TOF images are represented as a high dimensional tensor $\mathbf{T} \in \mathbb{R}^{(M,2048,2048)}$. These images are then further processed at a cell by cell basis into $\mathbf{Y} \in \mathbb{R}^{(M,64,64)}$ patches, where a cell is at the center of the patch along with its neighbors. Next, we construct semantic segmentation maps $\mathbf{S} \in \mathbb{R}^{(C+1,64,64)}$, where a vector $\mathbf{S}_{:,i,j}$ is one-hot encoded based on a cell type $C = 17$, and the $C + 1$-th channel denotes empty segmentation space. The data is train-test split at a 9:1 ratio at the MIBI-TOF image level to avoid cell neighborhood bias. We also use a synthetic test set where cells

---

[2]The dark green cell is extremely rare, classified as "other immune" and is a noise class label.

and their neighbors are sequentially modified to observe how varying cell type, position, and size affects the progressive changes in protein localizations[3].

## 4 METHODS

### 4.1 MODEL ARCHITECTURE

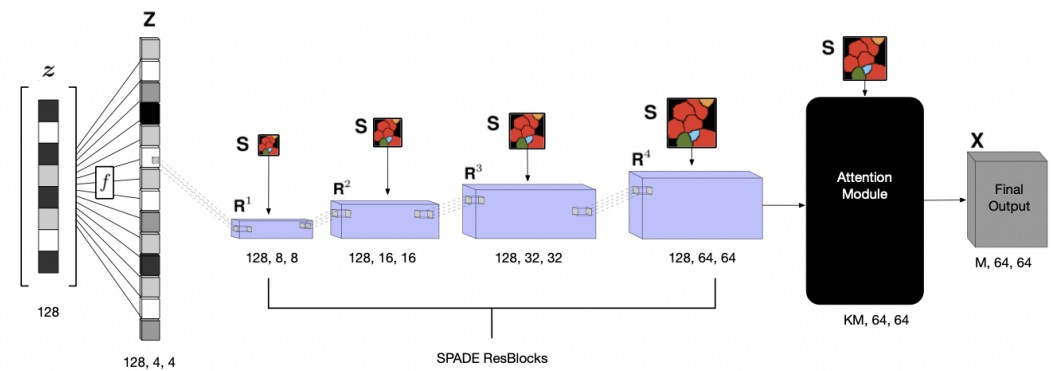

Figure 3: CCIGAN

We use SPADE residual blocks (Park et al., 2019) as our generative backbone and DCGAN's discriminator's architecture (Figure 3, A.1) (Radford et al., 2015). Park et al. (2019) have shown SPADE to be an effective way to inject conditioning into a generative model. The SPADE normalization layer serves as a replacement for previous layer normalization techniques. Instead of learning a universally shared per channel affine transformation, like in Batch Normalization (Ioffe & Szegedy, 2015) or Instance Normalization (Ulyanov et al., 2016), SPADE learns to predict affine transformations based on segmentation maps; each feature is uniquely transformed based on its cell type, size, and neighboring cells. The ability for SPADE to modulate activations based on the context of adjacent cell segmentations allows the network to effectively model the behaviors and interactions of cells. The input of CCIGAN is a noise vector $\boldsymbol{z} \in \mathbb{R}^{128}$ and a segmentation map $\mathbf{S}$. $f$ denotes a linear layer $\mathbb{R}^{128} \mapsto \mathbb{R}^{2048}$. $\mathbf{R}^i$ are feature map representations from SPADE resblocks and $\mathbf{X}$ denotes the final output of $M$ cell expressions. Below, each layer's output dimensions are given next to their respective equations.

$$\mathbf{Z} \in \mathbb{R}^{(128,4,4)} = f(\boldsymbol{z}) \tag{1}$$

$$\mathbf{R}^1 \in \mathbb{R}^{(128,8,8)} = \text{SPADE\_RESBLK}(\mathbf{Z}, \mathbf{S}) \tag{2}$$

$$\mathbf{R}^2 \in \mathbb{R}^{(128,16,16)} = \text{SPADE\_RESBLK}(\mathbf{R}^1, \mathbf{S}) \tag{3}$$

$$\mathbf{R}^3 \in \mathbb{R}^{(128,32,32)} = \text{SPADE\_RESBLK}(\mathbf{R}^2, \mathbf{S}) \tag{4}$$

$$\mathbf{R}^4 \in \mathbb{R}^{(128,64,64)} = \text{SPADE\_RESBLK}(\mathbf{R}^3, \mathbf{S}) \tag{5}$$

$$\mathbf{X} \in \mathbb{R}^{(M,64,64)} = \text{ATTENTION}(\mathbf{R}^4, \mathbf{S}) \tag{6}$$

### 4.2 ATTENTION MODULE

Our architectural contribution is a protein marker dependent attention module in the final output layer. The goal of the attention module is to condition the final output of a channel on a protein marker $m$ and $\mathbf{S}$'s cell types. For example the protein marker, pan-keratin $m_{\text{pk}}$, is expressed exclusively in tumor cells but not in other cells. Appropriately, an attention mechanism should attend to tumor cells and ignore irrelevant cells in $\mathbf{S}$ for $m_{\text{pk}}$. To replicate a marker searching for specific cell types that express it, we define a learned persistent vector for each marker denoted by $\boldsymbol{s}_{m \in M} \in \mathbb{R}^8$ that undergo a series of operations (Figure 4) with the final feature map representation attending to $m$'s specific cell types. It is also worthwhile to note that these persistent vectors $\boldsymbol{s}_m$ offer a degree of model interpretability that mimic real world markers. The current input dimensions to the attention module are $\mathbb{R}^{(128,64,64)}$ following the last resblock $\mathbf{R}^4$ and $m$ indexes from $1, .., M$.

---

[3]The test dataset will be released along with trained models for reproducibility after the reviewing period.

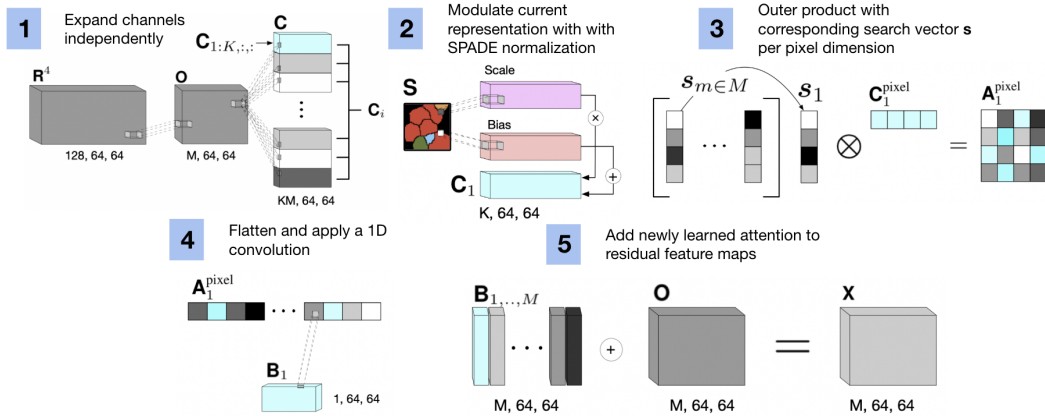

Figure 4: Attention module. This illustrates an instance, focusing on the light blue block $\mathbf{C}_1$.

$$\mathbf{O} \in \mathbb{R}^{(M,64,64)} = \text{CONV2D}(\mathbf{R}^4) \qquad (7)$$

$$\mathbf{C} \in \mathbb{R}^{(KM,64,64)} = \text{CONV2D}(\mathbf{O}) \qquad (8)$$

$$\mathbf{C}_i \in \mathbb{R}^{(K,64,64)} = \text{SPADE}(\mathbf{C}_{K(i-1):Ki,:,:}, \mathbf{S}) \quad (9)$$

$$\mathbf{A}_i \in \mathbb{R}^{(|\mathbf{s}| \times K,64,64)} = \mathbf{C}_i \otimes \boldsymbol{s}_m \qquad (10)$$

$$\mathbf{B}_i \in \mathbb{R}^{(1,64,64)} = \sigma(\text{CONV2D}(\mathbf{A}_i)) \quad (11)$$

$$\mathbf{X} \in \mathbb{R}^{(M,64,64)} = \mathbf{O} + \mathbf{B}_{1,..,M} \qquad (12)$$

Shown in Figure 4, after $\mathbf{R}^4$, a bottleneck convolution is applied to match the original data's dimension as $\mathbf{O}$ (step 1), which is used in a residual manner with the final output. Intuitively at this stage, $\mathbf{O}$'s feature maps resemble the target $\mathbf{Y}$, but we wish to further refine the output channels. We convolve $\mathbf{O}$ into $MK$ channeled features for each protein marker where $K = 8$. Considering each $\mathbf{C}_i$ where $i \in \{1, ..., M\}$ as a group of $K$ channels, the model spatially adaptive normalizes each $\mathbf{C}_i$ and computes an outer product with the corresponding persistent vector $\boldsymbol{s}_i$ and $\mathbf{C}_i$. The resulting matrix is flattened and convolved (with a kernel size of 1 on the pixel level) from $\mathbf{A}_i \in \mathbb{R}^{(|\mathbf{s}| \times K,64,64)} \mapsto \mathbb{R}^{(1,64,64)}$ followed by a sigmoid $\sigma(\cdot)$ activation. Lastly, the attentions $\mathbf{B}_{1,...,M}$ are added to $\mathbf{O}$ to obtain the output $\mathbf{X}$.

Initially, the model has no priors over the interaction of protein markers and cell types. The proposed outer product attention layer (outer product and $1 \times 1$ convolution) excels at modeling these relationships and interactions between specific markers and cell types. By using an outer product, the model forces attention at a pairwise pixel level comparison for all combinations of elements between $\boldsymbol{s}_m$ and $\mathbf{A}_i$. As training progresses, both the learned features over segmentation patches and the learned persistent vectors $\boldsymbol{s}_m$ improve, in turn allowing the learned $1 \times 1$ convolution to reason about positive or negative relationships from the pairwise pixel combinations.

### 4.3 IMPLEMENTATION DETAILS AND TRAINING REGIMEN

Our implementation (A.1) of the generator applies Spectral Norm to all layers (Miyato et al., 2018). The discriminator's input is the output of the generator concatenated with the segmentation patch $[\mathbf{X}, \mathbf{S}]$ and $[\mathbf{Y}, \mathbf{S}]$ for the ground truth. Finally CCIGAN uses ADAM ($lr_G = 0.0004$, $lr_D = 0.0001$) with GAN loss and feature matching loss. Full training details and loss functions are given in A.2.

## 5 EVALUATION

To conduct fair experiments, all models were optimized, tuned, and set with similar parameters. They were also taken from their official online implementations and trained for 120 epochs or until convergence (max 150). CCIGAN is identical to our designed SPADE comparison baseline with the exception of the attention module. Three experiments were conducted to validate the trained model's utility in generating biologically meaningful cellular proteins in the tumor microenvironment and ability to recapitulate and *quantify* previously established biological phenomena. Each subsection describes the experiment and the relevant metrics used in evaluation. Full mathematical definitions are given in section A.3.

## 5.1 IMAGE EVALUATION AND RECONSTRUCTION

First, we use the following evaluation metrics in order to compare with baseline results: adjusted $L_1$ and MSE score, $L_1$ and MSE score, structural similarity (SSIM) index (Wang et al., 2004) and cell based mutual information (MI) shown in Table 1. Equations and explanations are given in A.3.1. Bolded scores indicate the best scores.

| Metrics | CCIGAN | SPADE | Pix2PixHD | CycleGAN |
|---|---|---|---|---|
| Adjusted $L_1$ Score | **0.613** | **0.618** | 0.875 | 4.745 |
| $L_1$ Score | **0.594** | **0.602** | 0.745 | 3.959 |
| Adjusted MSE Score | **0.026** | 0.031 | 0.061 | 1.841 |
| MSE Score | **0.026** | 0.031 | 0.055 | 1.523 |
| SSIM | **0.810** | **0.802** | 0.709 | 0.394 |
| Cell Mutual Information | **10.46** | 10.25 | 9.26 | 7.96 |

Table 1: Comparison of conventional reconstruction metrics between different models.

| Experiment | Metrics | CCIGAN | SPADE | Pix2PixHD | CycleGAN |
|---|---|---|---|---|---|
| PD1-PDL1 (Tumor) | COM Score | **10.46** | 10.69 | 12.81 | 13.38 |
| | Random COM Score | 12.27 | 12.16 | 12.96 | 12.45 |
| PD1-PDL1 (Tumor) | EM Distance | **151.56** | -12.83 | -12.24 | -0.01 |
| | Positive EMD | **278.38** | 149.81 | 162.97 | 0.004 |
| | Projected EMD | **115.77** | 1.83 | 11.89 | -0.008 |
| | Random EMD | -123.43 | -83.11 | -12.50 | -0.006 |
| PD1-PDL1 (Endothelial) | EM Distance | -22.68 | -18.88 | -92.08 | 0.001 |
| | Positive EMD | 35.92 | 22.46 | 109.82 | 0.01 |
| | Projected EMD | -9.29 | -9.9 | -70.96 | 0.003 |
| Tumor-CD8 (Exp.) | $t$-test | **-13.81** | 2.68 | 25.20 | 14.27 |
| | $p$-value | $< 10^{-8}$ | 0.007 | $< 10^{-8}$ | $< 10^{-8}$ |
| Tumor-Tumor (Control) | $t$-test | 1.32 | 2.21 | 37.99 | 6.60 |
| | $p$-value | 0.187 | 0.027 | $< 10^{-8}$ | $< 10^{-8}$ |

Table 2: Biologically motivated experiments. Random, Endothelial, and Tumor-Tumor experiments are controls.

## 5.2 PD-1/PD-L1 RELATIONSHIPS BETWEEN T CELLS AND TUMOR CELLS

In the first step of biologically verifying the accuracy of CCIGAN, PD-1/PD-L1 expression relationships between CD8 T cells and tumor cells were assessed. Many T cells located within the tumor microenvironment have upregulated expression of PD-1 suggesting that the tumor milieu exerts influence on the protein localization and expression of infiltrating T cells (Ahmadzadeh et al., 2009). We assessed if increased expression of PD-L1 on neighboring tumor cells would result in increased directional PD-1 expression in a CD8 T cell (Figure 5). We also determined if there was a shift in the cell surface localization of the PD-1. To do so, we computed the Earth Mover's Distance between a CD8 T cell's PD-1 expression (represented as a histogram on a cell's polar coordinates) and itself before and after in different tumor scenarios. In addition to the mass, we computed the expected center of mass (COM) of PD-1 in a CD8 T cell with respect to the neighboring tumor PD-L1 expressions. Equations and explanations are given in A.3.2 and A.3.3. Additional figures illustrating this process are given in A.4.

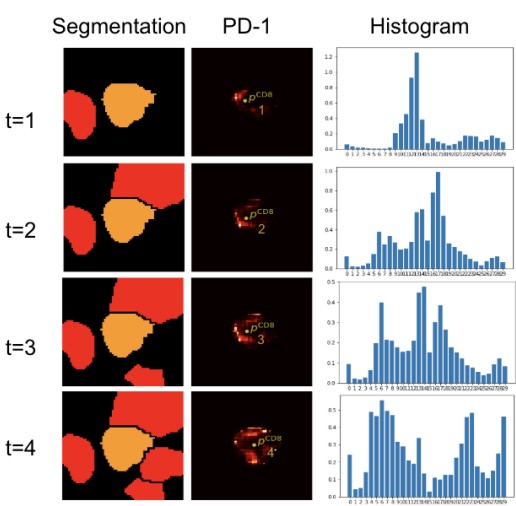

Figure 5: Example illustration of how a CD8 T cell's (orange) PD-1 histogram changes as a function of iteratively added tumor cells.

We expected that the properties of the PD-L1 expressing tumor cell would result in an increased directional PD-1 expression in a neighboring T cell. As a control, a PD-1 expressing T Cell was surrounded with endothelial cells (a normal tissue lining cell). As shown in Table 2, our initial hypothesis was confirmed in CCIGAN's predictions, wherein the presence of a PD-L1 expressing tumor cell, PD-1 expression in the CD8 T cell increased and moved towards the PD-L1 COM. On the other hand, the endothelial cell presence yielded no effect on the T cell's PD-1 expression, a result which is biologically expected. While both tumor and endothelial cells express PD-L1, the tumor microenvironment exercises more complex suppressive effects on CD8 T cells, which are more likely to modulate the T cell expression than endothelial cells. These findings verify CCIGAN captures the biological relationship of tumor cells inducing immunomodulatory changes on a neighboring T cell. None of the other models succeeded in capturing this protein relationship. This serves to highlight the capacity of CCIGAN to recapitulate established cell interaction phenomena within the tumor microenvironment.

### 5.3 Pan-keratin and CD8 Expression

Secondly, we assessed the effect of immune cell presence on tumor cell status markers. Keratins are a class of intracellular proteins that play an important role in ensuring cell structure. Furthermore, when CD8 T cells mediate tumor killing, they release enzymes which cleave the tumor cell's pan-keratin, disrupting the tumor cell structure (Oshima, 2002). We explored how the presence of neighboring CD8 T cells to a tumor cell would affect pan-keratin levels within the tumor cell, hypothesizing that T cell mediated tumor killing would result in a drop in pan-keratin tumor expression.

In this experiment, we used a Student's $t$-test as the statistical hypothesis test to evaluate the correlations between the pan-keratin expression in tumor cells and the area/number of CD8 T cells in contact with the tumor cell surface. Given a generated pan-keratin channel $\boldsymbol{X}_i \in \mathbb{R}^{(H,W)}$ and the segmentation map channel for CD8 T cells $\boldsymbol{S}_i \in \mathbb{R}^{(H,W)}$, we compute the total area of the cells $a_i = \sum_{h=1}^{H} \sum_{w=1}^{W} \boldsymbol{S}_i$, and the total expression level of pan-keratin $e_i = \sum_{h=1}^{H} \sum_{w=1}^{W} \boldsymbol{X}_i$. We then regress $\{e_i\}_{i=1}^{N}$ on $\{a_i\}_{i=1}^{N}$ and assess significance of the slope using a $t$-test against the null of no change in pan-keratin expression as a function of CD8 T cell-tumor contact.

CD8 T cells were placed adjacent to a tumor cell in increasing number, similar to Figure 1 (B)'s first row and Figure 5's experiment. It was observed that the presence of neighboring CD8 T cells to a tumor cell tended to decrease the pan-keratin expression in the tumor cell. This effect became more dramatic as the number of surrounding CD8 T cells was increased (A.5). As a control, when the tumor cell of interest was surrounded with adjacent tumor cell(s) instead of CD8 T cells, no change was noted in that tumor cell's pan keratin for CCIGAN. Table 2's fourth row shows the result of the $t$-test where the slope of the linear regression of pan-keratin expression with respect to the number of surrounding cells is compared to a $y = 0$ flat baseline. A statistically significant difference between the slope of the linear regression for the T cell scenario vs. the baseline was found in the main experiment, but was not present in the control scenario. These results indicate that the T cell presence mitigates a decrease in pan-keratin expression in the tumor cell, which is suggestive of T cell mediated tumor killing. Other models reported contrasting results where a significant change in the tumor cell pan-keratin expression was incorrectly reported under the control conditions. The resulting graphs are given in A.5. Further protein interaction patterns identified by CCIGAN are also reported in A.7. The findings from this experiment serve to highlight the robustness and accuracy of CCIGAN compared to existing image synthesis techniques in probing cell-cell interactions.

### 5.4 Tumor Infiltrated and Compartmentalized Microenvironments

As a final biological evaluation, we consider the variability of PD-L1 expression in immune and tumor cells across TNBC patient groups. Keren et al. (2018) determined that in situations of mixed tumor-immune architecture, where immune cells freely infiltrated the tumor, the tumor cells predominantly expressed PD-L1. Conversely, in situations of compartmentalized tumors, where there is a greater degree of delineation between immune and tumor cells, macrophages were the predominant source of expressed PD-L1, particularly at the tumor boundary. Figure 1 (B)'s row 2, 3 show examples of how directly manipulated segmentation maps can simulate the two microenvironments.

We used CCIGAN to predict on 200 directly manipulated mixed and non-mixed tumor environment segmentation patches. Similar to Section 5.3's experimental settings, we then compute the average expression of a specific marker for the cells of interest for all patches. For each experiment, we use endothelial cells as control cells to show our result has biological significance.

These findings were recapitulated by CCIGAN in Table 3. For a patient with a mixed tumor environment, when trained with mixed patient samples, CCIGAN reported increased PD-L1 expression on tumor cells. Furthermore, CCIGAN was able to quantify this difference in expression at the single cell level, reporting a tumor to macrophage PD-L1 expression ratio (bolded) of approximately 3.2 and 1.75 for patients A and B respectively.

| Experiment | Microenvironments | Patient A | Patient B |
|---|---|---|---|
| PD-1 (**T cell**) | T cell / Tumor / Macrophages | 0.01886 | 0.00131 |
| | T cell / Endothelial / Macrophages | 0.00558 | 0.00107 |
| PD-L1 (**Tumor**) | T cell / Tumor / Macrophages | **0.00649** | **0.00100** |
| | Endothelial / Tumor / Macrophages | 0.00279 | 0.00046 |
| PD-L1 (**Macrophages**) | T cell / Tumor / Macrophages | **0.00204** | **0.00057** |
| | T cell / Endothelial / Macrophages | 0.00068 | 0.00047 |

Table 3: Average PD-1/PD-L1 expression on the mixed tumor environment. The bolded cells indicate which cells are being measured.

Conversely, when trained with compartmentalized patient samples, CCIGAN reported increased PD-L1 expression on macrophages adjacent to tumor cells as compared to macrophages adjacent to normal endothelial (inert) cells for patients C and D. This difference was quantified as a ratio of PD-L1 expression of tumor-adjacent macrophages to endothelial-adjacent macrophages, approximately 1.85 and 2.7 for patient C and patient D respectively. Moreover, using the trained compartmentalized model to predict on mixed segmentation patches, CCIGAN still reports a 26% (patient C) and 19% (patient D) increase of macrophage PD-L1 expression when compared to mixed microenvironments (Table 4).

| Experiment (**Macrophages**) | Microenvironments | Patient C | Patient D |
|---|---|---|---|
| PD-L1 (Compartmentalized) | T cell / Tumor / Macrophages | 0.00408 | 0.00608 |
| | T cell / Endothelial / Macrophages | 0.00220 | 0.00225 |
| PD-L1 (Mixed) | T cell / Tumor / Macrophages | 0.00324 | 0.00510 |

Table 4: Average PDL1 expression of macrophages/monocytes on the compartmentalized tumor environment.

The PD-L1 ratios for the above two scenarios indicate that CCIGAN has appropriately captured previously reported biological outcomes and is capable of quantifying these phenomena at single cell levels. Furthermore, the model is adaptable to various different types of tumor architecture depending on its training set to produce different hypothesis testing environments. The agreement between CCIGAN data and those reported in Keren et al. (2018) serve as an important control and demonstrate the fidelity of the CCIGAN output towards true biological results. More importantly, this provides strong evidence to support the accuracy of the predictions made by CCIGAN in the assessment of hypothetical cellular scenarios which cannot be tested via *in vitro* tissue study alone.

## 5.5 MODEL INTERPRETABILITY

Examining the model's persistent vectors $s_m$, we can try to understand if there is a match between real world protein markers and the representations of $s_m$. For example, the vector $s_{\text{pk}}$ for pan-keratin attends to tumor cells and $s_{\text{CD8}}$ attends to CD8 T cells at pixel pairwise levels. It follows that in a simple experiment where corresponding $s_{\text{CD8}} \leftrightarrow s_{\text{pk}}$ vectors are exchanged internally in the attention module (Eq. 10, Figure 4 Step 3, outer product) we may observe a lower expression for tumor cells in channel $m_{\text{pk}}$ and a lower expression for CD8 T cells in channel $m_{\text{CD8}}$ since tumor cells do not express CD8 and CD8 T cells do not express pan-keratin. As a control, we also switch surface membrane markers HLA Class 1 and dsDNA markers as they are present in all cells and have very similar average expression values ($s_{\text{HLAc1}} \leftrightarrow s_{\text{dsDNA}}$). Accordingly, for our control, we expect to see negligble changes. We define the expression ratio as $\frac{\text{after}}{\text{before}} - 1$.

| Protein Markers | CD8 | pan keratin | HLA Class 1 | dsDNA |
|---|---|---|---|---|
| Expression Ratios | -0.373 | -0.145 | -0.054 | -0.0012 |

Table 5: $s_m$ persistent vector interpretability experiments.

In Table 5, we can see a larger magnitude decrease of the expression ratios in the $s_{\mathrm{CD8}} \leftrightarrow s_{\mathrm{pk}}$ experiment and a minute difference in the $s_{\mathrm{HLAc1}} \leftrightarrow s_{\mathrm{dsDNA}}$. Further visualizations (Figure 13) and discussion (model generativeness, Figure 14) are given in A.6.

## 6 Conclusion and Discussion

We introduced the idea of applying image synthesis to understanding and exploring cell-cell interactions in various and different contexts. To do so we use a protein attention based GAN, CCIGAN, which can provide accurate characterizations of cellular protein localization phenomena from conditioned counterfactual cell-cell scenarios. Additionally, the architecture of the attention module we propose can be generalized to other multiplexed datasets that require real world priors.

Furthermore, CCIGAN outperforms a variety of current methods in biological modeling. We demonstrate this through biological consistency where CCIGAN recapitulates, discovers, and quantifies meaningful cellular interactions through 3 different experiments in a tumor environment unrecognized by other models. This highlights the potential for CCIGAN to identify cellular protein interactions which account for variation in patient responses to cancer therapy, providing a framework for biological hypotheses which explain clinical outcomes on a cellular level.

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

## A    APPENDIX

### A.1    ARCHITECTURE DETAILS

The detailed architecture of our generator is shown on Table 6.

| Layers | Output Size | Generator |
|---|---|---|
| Linear | $(128, 4, 4)$ | Linear $128 \times 2048$ |
| Upsampling | $(128, 8, 8)$ | Upsampling $2 \times 2$ |
| SPADE ResBlk-1 | $(128, 8, 8)$ | SPADE 128, Leaky ReLU
Convolution $3 \times 3$
SPADE 128, Leaky ReLU
Convolution $3 \times 3$ |
| Upsampling | $(128, 16, 16)$ | Upsampling $2 \times 2$ |
| SPADE ResBlk-2 | $(128, 16, 16)$ | SPADE 128, Leaky ReLU
Convolution $3 \times 3$
SPADE 128, Leaky ReLU
Convolution $3 \times 3$ |
| Upsampling | $(128, 32, 32)$ | Upsampling $2 \times 2$ |
| SPADE ResBlk-3 | $(64, 32, 32)$ | SPADE 128, Leaky ReLU
Convolution $3 \times 3$
SPADE 64, Leaky ReLU
Convolution $3 \times 3$
SPADE 64, Leaky ReLU
Shortcut Convolution $3 \times 3$ |
| Upsampling | $(64, 64, 64)$ | Upsampling $2 \times 2$ |
| SPADE ResBlk-4 | $(64, 64, 64)$ | SPADE 64, Leaky ReLU
Convolution $3 \times 3$
SPADE 64, Leaky ReLU
Convolution $3 \times 3$ |
| Convolution | $(24, 64, 64)$ | Leaky ReLU, Convolution $5 \times 5$ |
| Convolution | $(24 * 8, 64, 64)$ | Leaky ReLU, Convolution $5 \times 5$ |
| Group SPADE | $(24 * 8, 64, 64)$ | [SPADE 8] * 24 |
| Modulation | $(24 * 64, 64, 64)$
$(24, 64, 64)$ | [Outer Product $8 \otimes 8$] * 24
Convolution $1 \times 1$, Sigmoid |
| Output | $(24, 64, 64)$ | Sum residual, Sigmoid |

Table 6: Architecture details of CCIGAN's generator

where ResBlk is the residual block with skip connection used in ResNet (He et al., 2016), and SPADE is the spatially-adaptive normalization layer. The detailed architecture of our discriminator is shown on Table 7.

| Layers | Output Size | Discriminator |
|---|---|---|
| Conv-1 | $(32, 32, 32)$ | Convolution $4 \times 4$, stride 2
Instance Norm, Leaky ReLU |
| Conv-2 | $(64, 16, 16)$ | Convolution $4 \times 4$, stride 2
Instance Norm, Leaky ReLU |
| Conv-3 | $(128, 8, 8)$ | Convolution $4 \times 4$, stride 2
Instance Norm, Leaky ReLU |
| Conv-4 | $(256, 4, 4)$ | Convolution $4 \times 4$, stride 2
Instance Norm, Leaky ReLU |
| Conv-5 | $(512, 2, 2)$ | Convolution $4 \times 4$, stride 2
Instance Norm, Leaky ReLU |
| Conv-6 | $(1, 1, 1)$ | Convolution $3 \times 3$, stride 2
Sigmoid |

Table 7: Architecture details of CCIGAN's discriminator

For all baseline models, we use the architecture based on the their original implementation. Due to the size of the cell patch is $(64, 64)$, we reduce the size of hidden layers to fit our dataset. For fair comparison, we use the same reduction of hidden layers and the same discriminator architecture for SPADE, pix2pixHD, and CCIGAN.

## A.2 MODEL TRAINING

$G$ is the generator and $D$ is the discriminator for CCIGAN. Given segmentation map **S**, ground truth **Y** and noise $\delta$, the generated image is $\mathbf{X} = G(\mathbf{S}, \delta)$. The input of the discriminator is the cell image conditioned on the segmentation map **S**. We use LSGAN loss (Mao et al., 2017) in CCIGAN, which is defined as follows:

$$L_{GAN}(G, D) = \mathbb{E}_{\mathbf{Y}, \mathbf{S}} \left[ \|D(\mathbf{Y}, \mathbf{S})\|_2 \right] + \mathbb{E}_{\mathbf{S}} \left[ \|1 - D(G(\mathbf{S}, \delta), \mathbf{S})\|_2 \right] \tag{13}$$

In addition to GAN loss, we also use feature matching loss (Wang et al., 2018a) during training expressed as:

$$L_{FM}(G, D) = \mathbb{E}_{\mathbf{Y}, \mathbf{S}} \sum_{j=1}^{J} \frac{1}{N_j} \left[ \|D_j(\mathbf{Y}, \mathbf{S}) - D_j(G(\mathbf{S}, \delta), \mathbf{S})\|_1 \right] \tag{14}$$

where $D_j$ is $j$-th layer feature map of the discriminator for $j \in \{1, ..., J\}$, and $N_j$ is the number of elements in $j$-th layer. Consequently, the objective function for training is given as follows:

$$\min_G \left( \left( \max_D L_{GAN}(G, D) \right) + \lambda L_{FM}(G, D) \right) \tag{15}$$

where $\lambda = 10$. Due to the size of cell patch is $(64, 64)$, we do not use multi-scale discriminators and perceptual loss in CCIGAN and other baseline models e.g. SPADE and pix2pixHD.

In training, we use ADAM as the optimizer. The generator learning rate is $lr_G = 0.0004$ and the discriminator learning rate is $lr_D = 0.0001$. We train CCIGAN 120 epochs with a training set of 5648 cell patches. We train other baseline models for 120 epochs or until they converge (max 150). The full details of training of CCIGAN and baselines are shown as Table 8. The hyperparameters of each model are fine-tuned to get better performance. The training time was roughly equal for all models. In particular, CCIGAN was around 1.2 times slower than the SPADE baseline on a single Tesla V100 GPU.

| Metrics | Ours | SPADE | Pix2PixHD | CycleGAN |
|---------|------|-------|-----------|----------|
| $lr_G$ | 0.0004 | 0.0008 | 0.0002 | 0.0002 |
| $lr_D$ | 0.0001 | 0.0001 | 0.0002 | 0.0002 |

Table 8: Hyperparameters of models

## A.3 EVALUATION METRICS

### A.3.1 RECONSTRUCTION METRICS

Given the generated image set $\mathcal{X} = \{\mathbf{X}_i\}_{i=1}^N$ and the ground truth set $\mathcal{Y} = \{\mathbf{Y}_i\}_{i=1}^N$ with $\mathbf{X}_i, \mathbf{Y}_i \in \mathbb{R}^{(M, H, W)}$, the $L_1$/MSE score is defined as follows,

$$L(\mathcal{X}, \mathcal{Y}) = \sum_{i=1}^{N} \sum_{m=1}^{M} \|\text{sort}(\boldsymbol{U}_i \odot \mathbf{X}_{i,m}) - \text{sort}(\boldsymbol{U}_i \odot \mathbf{Y}_{i,m})\|_* \tag{16}$$

where $\|\cdot\|_*$ can be either $L_1$ or $L_2$ norm, $\odot$ is the element-wise product, $\mathbf{X}_{i,m}$ and $\mathbf{Y}_{i,m}$ are the $m$-th channel of the $i$-th cell patch, $\boldsymbol{U}_i \in \{0, 1\}^{(H, W)}$ is the mask matrix which masks all the cells in $i$-th patch. For any matrix $\boldsymbol{A}$, $\text{sort}(\boldsymbol{A})$ is the sort function that sorts all entries of $\boldsymbol{A}$. The sorting function ensures our metrics are position independent and only measures the intensity of the generated image and ground truth. The score function $L(\mathcal{X}, \mathcal{Y})$ only computes the loss of sorted expression inside of

the cells. Then we add penalization for expression outside of cells. The adjusted $L_1$/MSE score is introduced as follows,

$$L_{\text{adj}}(\mathcal{X}, \mathcal{Y}) = \sum_{m=1}^{M} \Big( \|\text{sort}(\boldsymbol{U}_i \odot \mathbf{X}_{i,m}) - \text{sort}(\boldsymbol{U}_i \odot \mathbf{Y}_{i,m})\|$$
$$- \|\text{sort}((\mathbf{1}_d - \boldsymbol{U}_i) \odot \mathbf{X}_{i,m}) - \text{sort}((\mathbf{1}_d - \boldsymbol{U}_i) \odot \mathbf{Y}_{i,m})\| \Big) \tag{17}$$

where $\mathbf{1}_d$ is the matrix with all entries equal to 1. The (adjusted) $L_1$/MSE scores of CCIGAN and baseline models are shown on Table 1. A smaller score means a better result.

For any two images $\boldsymbol{X}, \boldsymbol{Y} \in [0,1]^{(H,W)}$, the SSIM and MI are defined as:

$$\text{SSIM}(\boldsymbol{X}, \boldsymbol{Y}) = \frac{(2\mu_X \mu_Y + c_1)(2\sigma_{XY} + c_2)}{(\mu_X^2 + \mu_Y^2 + c_1)(\sigma_X^2 + \sigma_Y^2 + c_2)} \tag{18} \qquad I(\boldsymbol{X}; \boldsymbol{Y}) = H(\boldsymbol{X}) + H(\boldsymbol{Y}) - H(\boldsymbol{X}, \boldsymbol{Y}) \tag{19}$$

where $H(\cdot)$ is entropy, $\mu_X$ and $\sigma_X$ are the mean and standard deviation of $\mathbf{X}$, $c_1, c_2$ are constants. Then the SSIM between $\mathcal{X}, \mathcal{Y}$ is

$$\text{SSIM}(\mathcal{X}, \mathcal{Y}) = \frac{1}{N} \frac{1}{M} \sum_{i=1}^{N} \sum_{m=1}^{M} \text{SSIM}(\mathbf{X}_{i,m}, \mathbf{Y}_{i,m}) \tag{20}$$

In cell based MI, test patches are processed at a cell-cell basis where their mutual information is computed with the corresponding cell in the ground truth. For the generated image $\mathbf{X}_i$ of the $i$-th patch, we assume there are $T_i$ cells in the $i$-th patch. Then for each cell $t$, the pixels of $m$-th channel of the $t$-th cell in the $i$-th patch can be expressed as a vector $\boldsymbol{x}_{i,m}^t$. Hence, the cell based MI is formulated as:

$$I(\mathcal{X}; \mathcal{Y}) = \frac{1}{\sum_{i=1}^{N} T_i} \frac{1}{M} \sum_{i=1}^{N} \sum_{m=1}^{M} \left( \sum_{t=1}^{T_i} I(\boldsymbol{x}_{i,m}^t; \boldsymbol{y}_{i,m}^t) \right) \tag{21}$$

We report $I(\mathcal{X}; \mathcal{Y})$ on Table 1. The SSIM measures the similarity between the generated image and the ground truth. For SSIM, we use HLA Class 1 and dsDNA due to the their expressions in all cells. If all channels were considered, the SSIM would be uninformative due to the majority of the channels being blank or sparse. The MI measures the information shared between generated image and ground truth at a cell by cell basis where we consider all channels. Consider the example where a model generates no expression in marker $m$ but the real data has expression in $m$, the MI would be 0 and vice versa. Higher SSIM and MI values mean better results. Table 1 demonstrates that CCIGAN outperforms or matches all other baselines on all reconstruction metrics.

### A.3.2 Weighted Centroid

For a generated cell image, its centroid is the mean position of all the points in the cell. We use pixel values as weights in computing the weighted centroid, now referred to as center of mass (COM). Given a cell image $\boldsymbol{X} \in \mathbb{R}^{(H,W)}$, with indices of the segmented cell $V \subseteq \{1, \ldots, H\} \times \{1, \ldots, W\}$, the COM $\bar{\boldsymbol{p}} = (\bar{x}, \bar{y})$ is defined as $\bar{x} = \frac{\sum_{(x,y) \in V} x \boldsymbol{X}_{x,y}}{\sum_{(x,y) \in V} \boldsymbol{X}_{x,y}}$ and $\bar{y} = \frac{\sum_{(x,y) \in V} y \boldsymbol{X}_{x,y}}{\sum_{(x,y) \in V} \boldsymbol{X}_{x,y}}$.

In the PD-1/PD-L1 experiment, we compute the COM of the CD8 T cell (cell of interest) weighted by PD-1 expression, given as $\bar{\boldsymbol{p}}^{\text{CD8}}$, and the COM of all tumor cells weighted by PD-L1 expression, given as $\bar{\boldsymbol{p}}^{\text{Tumor}}$. Since T cells located within the tumor microenvironment often have upregulated expression of PD-L1, we assume that $\bar{\boldsymbol{p}}^{\text{CD8}}$ should have the same COM as all of its surrounding tumor cells $\bar{\boldsymbol{p}}^{\text{Tumor}}$. The center of mass score is defined below as the relative distance between $\bar{\boldsymbol{p}}^{\text{CD8}}$ and $\bar{\boldsymbol{p}}^{\text{Tumor}}$, where $N$ is defined as the number of patches:

$$\text{COM}_{\text{projection}} = \frac{1}{N} \sum_{i=1}^{N} \|\bar{\boldsymbol{p}}_i^{\text{CD8}} - \text{Proj}_{\text{CD8}}(\bar{\boldsymbol{p}}_i^{\text{Tumor}})\|_2 \tag{22}$$

The projection function $\text{Proj}(\cdot)$ is used to project $\bar{\boldsymbol{p}}^{\text{Tumor}}$ onto the CD8 T cell to ensure the expected COM of the tumor cells is inside of the CD8 T cell. As a reference we choose a random position $\bar{\boldsymbol{p}}^{\text{Random}}$ in the CD8 T cell (PD-1) which replaces $\bar{\boldsymbol{p}}^{\text{CD8}}$ in Eq. 22 and compute the random COM score to show the effectiveness of the result. An example illustration is given in Figure 6 A.4.

### A.3.3 EARTH MOVER'S DISTANCE

For each segmentation map $i$, we iteratively add its $T_i$ tumor cells around one CD8 T cell. The COM is defined for the $t$-th tumor cell as $\bar{\boldsymbol{p}}_{t,i}^{\text{Tumor}}$ for $t \in \{1, ..., T_i\}$. We omit the subscript $i$ when it is clear from context. The subsequent instances of the PD-1 COMs in the CD8 T cell by adding the $t$-th tumor are given by $\bar{\boldsymbol{p}}_t^{\text{CD8}}$. Initially when there are no tumor cells, we define $\bar{\boldsymbol{p}}_0^{\text{CD8}}$ as the centroid of the CD8 T cell. Based on the above setting, we define a vector $\boldsymbol{v}_t$ which points from the centroid of the CD8 T cell $\bar{\boldsymbol{p}}_0^{\text{CD8}}$, to the COM of the $t$-th tumor cell $\bar{\boldsymbol{p}}_t^{\text{Tumor}}$. We define vector $\boldsymbol{u}_t$ which points from the previous COM $\bar{\boldsymbol{p}}_{t-1}^{\text{CD8}}$ to the current COM $\bar{\boldsymbol{p}}_t^{\text{CD8}}$ of the CD8 T cell. We define $\theta_t$ as the angle between $\boldsymbol{v}_t, \boldsymbol{u}_t$. If $\cos\theta_t > 0$, that is to say if the cosine similarity is positive, the COM of a CD8 T cell $\bar{\boldsymbol{p}}_t^{\text{CD8}}$, moves correctly towards the COM of the added tumor cell $\bar{\boldsymbol{p}}_t^{\text{Tumor}}$. An illustration of the points and vectors is given in Figure 8. Formally:

$$\boldsymbol{v}_t = \bar{\boldsymbol{p}}_t^{\text{Tumor}} - \bar{\boldsymbol{p}}_0^{\text{CD8}}, \; \boldsymbol{u}_t = \bar{\boldsymbol{p}}_t^{\text{CD8}} - \bar{\boldsymbol{p}}_{t-1}^{\text{CD8}}, \; \cos\theta_t = \frac{\boldsymbol{u}_t \cdot \boldsymbol{v}_t}{\|\boldsymbol{u}_t\| \cdot \|\boldsymbol{v}_t\|} \tag{23}$$

After obtaining the directional information, we use Earth Mover's Distance (EMD) (Rubner et al., 2000) to evaluate the changes in PD-1 expression of the CD8 T cell. The EMD, which measures the dissimilarity of two distributions, is used in this context to measure the protein localization shifts in PD-1 before and after adding a tumor cell. We consider each cell $\boldsymbol{X}$ in polar coordinates $(r, \theta)$ with respect to its centroid, integrate its expression along the radius coordinates, and evaluate the resulting histogram $\text{hist}(\boldsymbol{X})$ along the angle coordinate. This allows for the definition of distance for moving one histogram to another, i.e. $\text{em}(\boldsymbol{X}_i^t, \boldsymbol{X}_i^{t-1}) = d_{\text{EM}}(\text{hist}(\boldsymbol{X}_i^t), \text{hist}(\boldsymbol{X}_i^{t-1}))$, for the generated PD-1 expression of the CD8 T cell $\boldsymbol{X}_i^t$ when adding the $t$-th tumor cell. The final EMD score is defined as:

$$\text{EMD} = \frac{1}{\sum_{i=1}^N T_i} \sum_{i=1}^N \sum_{t=1}^{T_i} \mathbb{1}(\|\boldsymbol{X}_i^t\| > \|\boldsymbol{X}_i^{t-1}\|) \cdot \text{em}(\boldsymbol{X}_i^t, \boldsymbol{X}_i^{t-1}) \cdot \cos\theta_{t,i} \tag{24}$$

where the indicator function $\mathbb{1}(\cdot) = 1$ if and only if $\|\boldsymbol{X}_i^t\| > \|\boldsymbol{X}_i^{t-1}\|$, otherwise $\mathbb{1}(\cdot) = 0$. This ensures that the biological constraint of PD-1 expression increasing as a response to added tumor cells is met. Recall, if $\cos\theta_t > 0$, $\bar{\boldsymbol{p}}_t^{\text{CD8}}$ has moved in the direction of $\bar{\boldsymbol{p}}_t^{\text{Tumor}}$, implying the shift in PD-1 expression is correct, and in turn increases the EMD. By contrast, the EMD score decreases when $\bar{\boldsymbol{p}}_t^{\text{CD8}}$ moves in the opposite direction. Example figures and illustrations showing this process are given in Figure 7 A.4. Using the EMD we define a randomized search algorithm for discovering other cell-cell interactions; their results and discussion are given in A.7.

Based on the definition of EMD score, the positive EMD score is defined as:

$$\text{EMD}_{\text{positive}} = \frac{1}{\sum_{i=1}^N T_i} \sum_{i=1}^N \sum_{t=1}^{T_i} \mathbb{1}(\|\boldsymbol{X}_i^t\| > \|\boldsymbol{X}_i^{t-1}\|) \cdot \text{em}(\boldsymbol{X}_i^t, \boldsymbol{X}_i^{t-1}) \cdot \max\{\cos\theta_{t,i}, 0\} \tag{25}$$

The positive EMD score only evaluates the change in PD-1 expression when the COM of a CD8 T cell moves towards the COM of the added tumor cell. The projected EMD score is defined as:

$$\text{EMD}_{\text{projected}} = \frac{1}{\sum_{i=1}^N T_i} \sum_{i=1}^N \sum_{t=1}^{T_i} \mathbb{1}(\|\boldsymbol{X}_i^t\| > \|\boldsymbol{X}_i^{t-1}\|) \cdot \text{em}(\boldsymbol{X}_i^t, \boldsymbol{X}_i^{t-1}) \cdot \|\boldsymbol{u}_{t,i}\| \cos\theta_{t,i} \tag{26}$$

The projected EMD score is the EMD score weighted by $\|\boldsymbol{u}_{t,i}\|$, i.e. the shift from the previous COM to the current COM of the CD8 T cell.

### A.4 EVALUATION VISUALIZATIONS

Here we provide some example visualizations and illustrations center of mass nomenclature, mass movement of PD-1 as a function of neighboring PD-L1, and process of computing EM distance.

We can observe in Figure 7 that the mass inside of the T cell in the PD-1 channel shifts as a response to surrounding tumor cell expressions of PD-L1. The surrounding tumor PD-L1 expressions $\bar{\boldsymbol{p}}_t^{\text{Tumor}}$ are shown in the third row on a cell by cell basis for $t \in \{1, ..., T_i = 4\}$. Note that the 3rd column in PD-L1 has sparse expression. Finally the last row shows the PD-1 and PD-L1 channels superimposed into one channel.

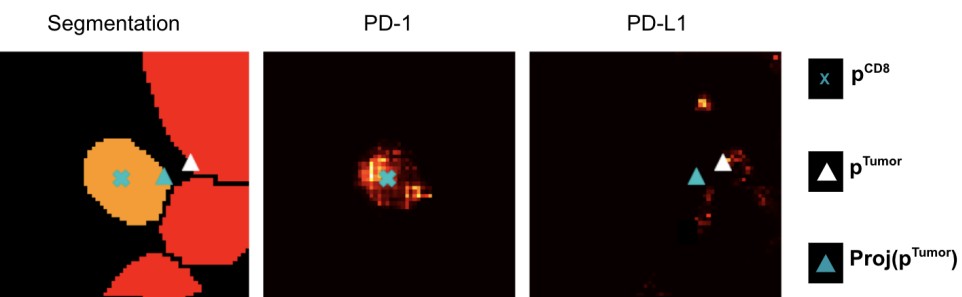

Figure 6: An example illustration of the center of mass (COM) nomenclature from section 5.1. Note the projection onto the CD8 T cell. This provides a more consistent measurement across different patches by projecting $p^{\text{Tumor}}$ onto the CD8 T cell.

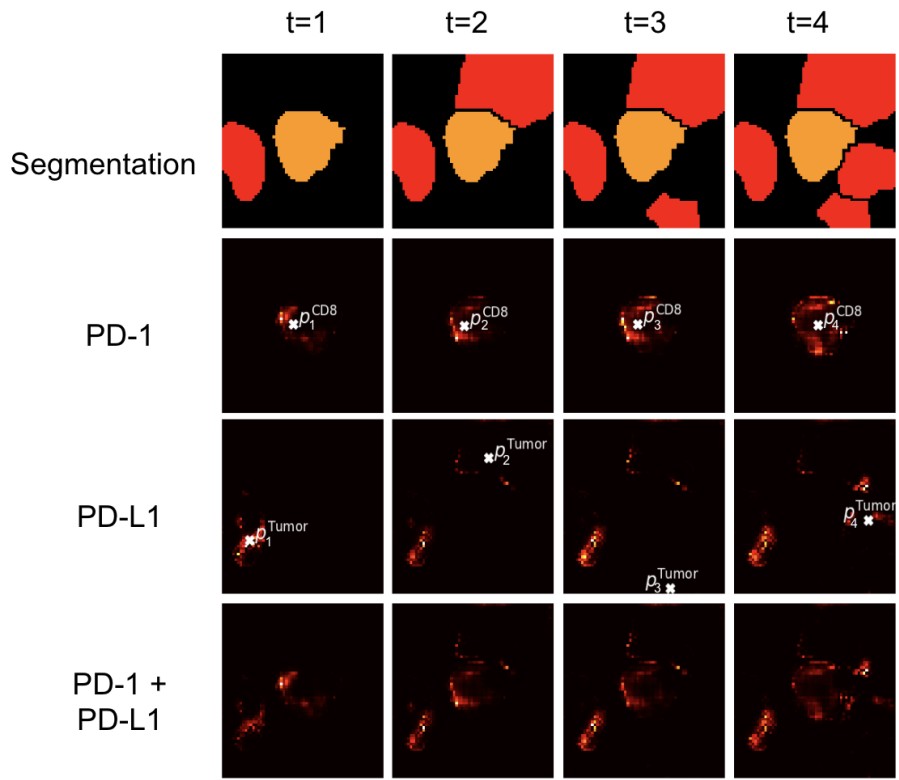

Figure 7: Process of iteratively adding tumor cells. The added red cells are tumor cells (PD-L1) and the center orange cell indicates a CD8 T cell (cell of interest, PD-1). For this process, we focus on each instance of an added tumor.

We give an example in Figure 8 to illustrate the vectors $v_t$ and $u_t$ after adding $t$-th tumor cell for $t = 1, 2$ in computing EMD score, where $v_t$ and $u_t$ defined in Eq. 23.

## A.5 RESULTS GRAPHS

The pan-keratin/CD8 experiment is similar to Figure 7's orientation except the center cell (cell of interest) is a tumor cell (red) and the adjacent neighboring cells are CD8 T cells (orange).

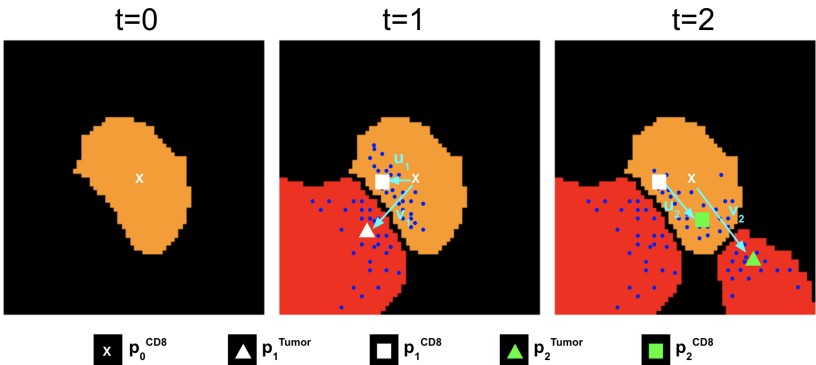

Figure 8: An example illustration of the points and vector nomenclature from section 5.2. The blue dots are the expression of PD-1 and PD-L1 proteins. The cyan arrows show the vectors $v_t$ and $u_t$. Note the shift in expression of the PD-1 as a response to the added tumor's PD-L1 expression.

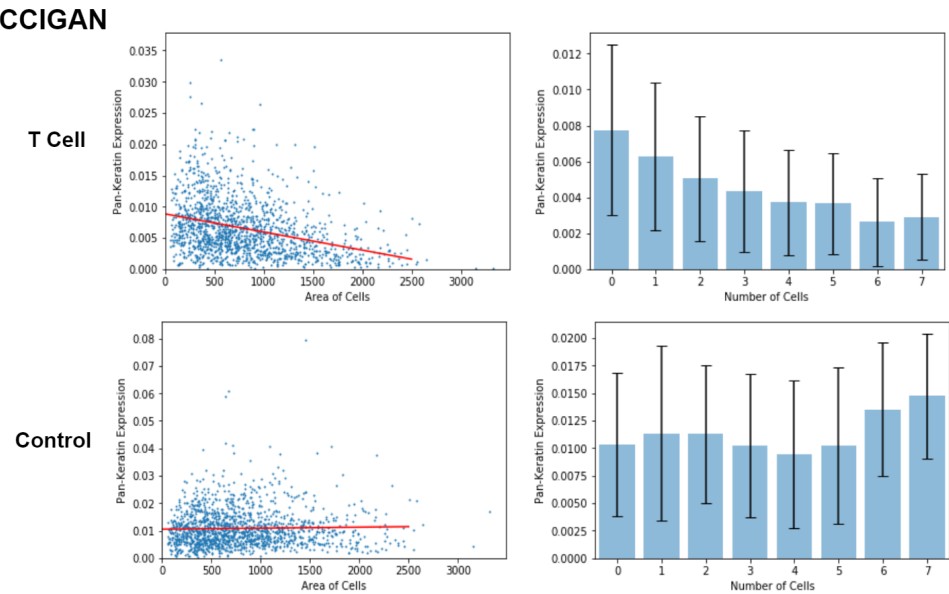

Figure 9: CCIGAN experiment for adding CD8 T cells and tumor cells (control) around a tumor cell.

CCIGAN predicted a decrease in tumor cell pan-keratin expression with respect to increasing CD8 T cell area/number (Figure 9). This is juxtaposed to the tumor cell control where there is no change in the pan-keratin level as the number of neighboring tumor cells is increased.

SPADE does not predict a decrease in tumor cell pan-keratin expression with respect to increasing CD8 T cell area/number and shows no difference in pan-keratin expression trends between the T cell and control groups (Figure 10).

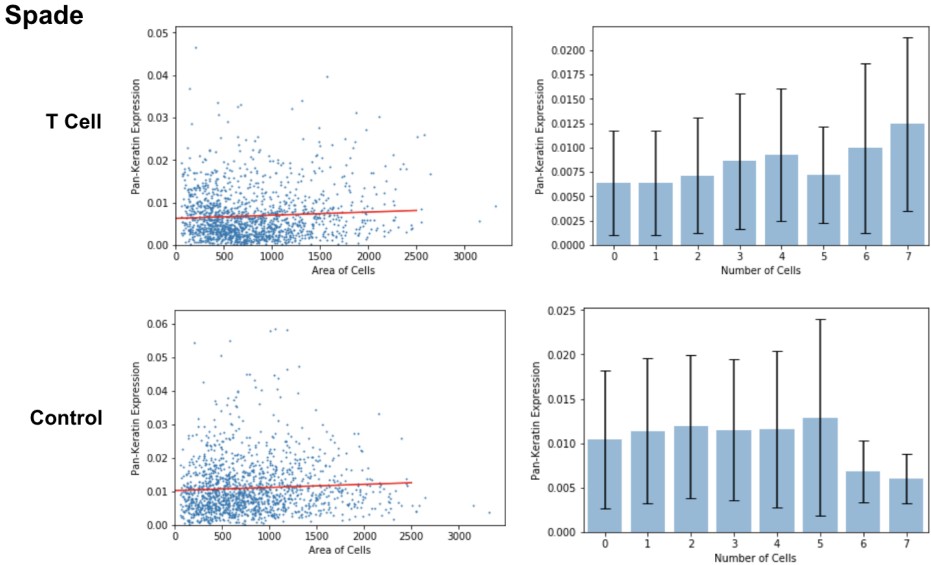

Figure 10: SPADE experiment for adding CD8 T cells and tumor cells (control) around a tumor cell.

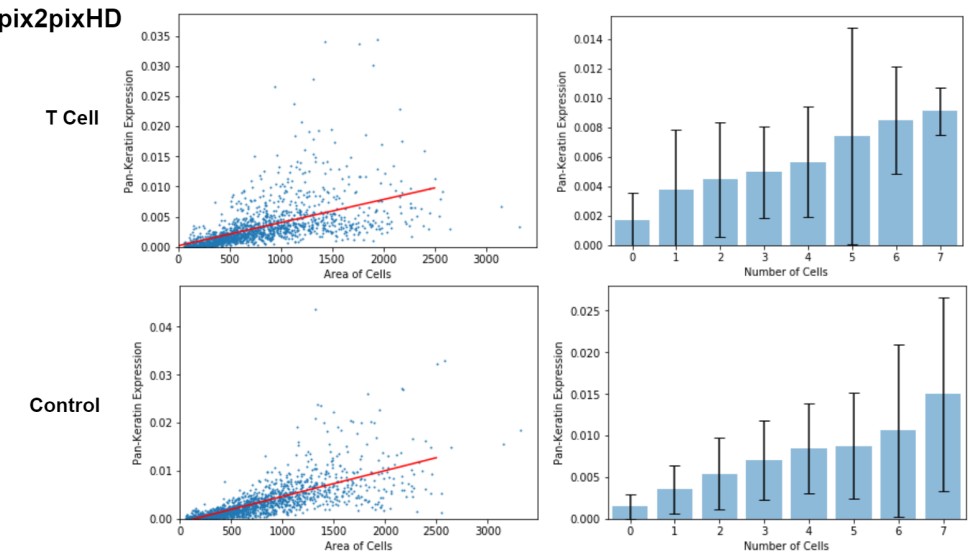

Figure 11: pix2pixHD experiment for adding CD8 T cells and tumor cells (control) around a tumor cell.

pix2pixHD erroneously predicts an increase in tumor cell pan-keratin expression with respect to increasing CD8 T cell area/number and shows no difference in pan-keratin expression trends between the T cell and control groups (Figure 11).

CycleGAN fails to predict a decrease in tumor cell pan-keratin expression with respect to increasing CD8 T cell area/number and shows no difference in pan-keratin expression trends between the T cell and control groups (Figure 12).

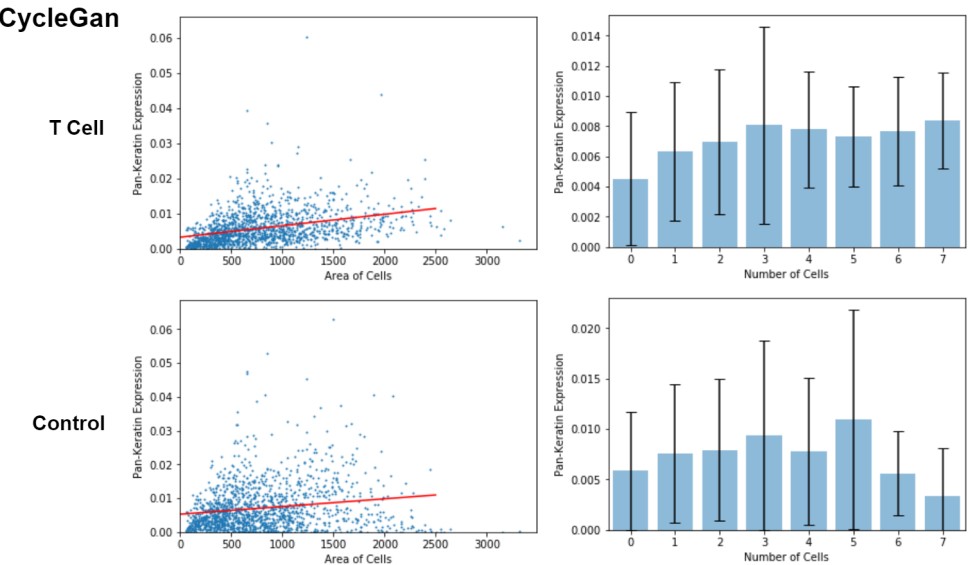

Figure 12: CycleGAN experiment for adding CD8 T cells and tumor cells (control) around a tumor cell.

### A.6 MODEL INTERPRETABILITY AND GENERATIVENESS

Figure 13 shows the persistent vectors $s_i$ for all proteins. Note the similarity between CD3 and CD8 T cell protein markers and the similarity between dsDNA and HLA Class 1 surface membrane proteins (expressed in all cells). It is also important to make the distinction that sparse markers (while different) are similar in state. This is due to the lack of training data for rare cell types, making it difficult for the model to reason on such a small sample size.

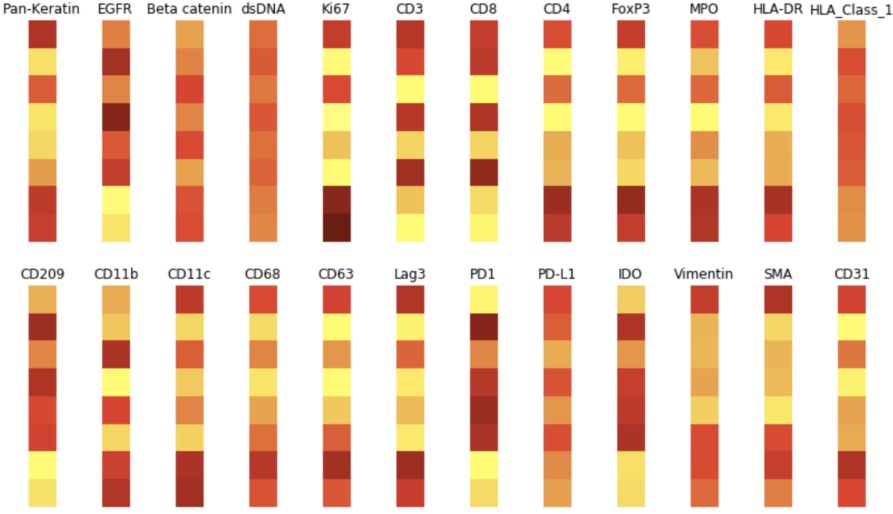

Figure 13: Persistent vectors $s$ for various channels.

Figure 14 shows the generativeness of CCIGAN through an uncertainty map over 100 instances (random noise). An uncertainty map shows the differences per pixel $(x, y)$ location. The higher intensities indicate a higher probability of changing at the specified $(x, y)$ location.

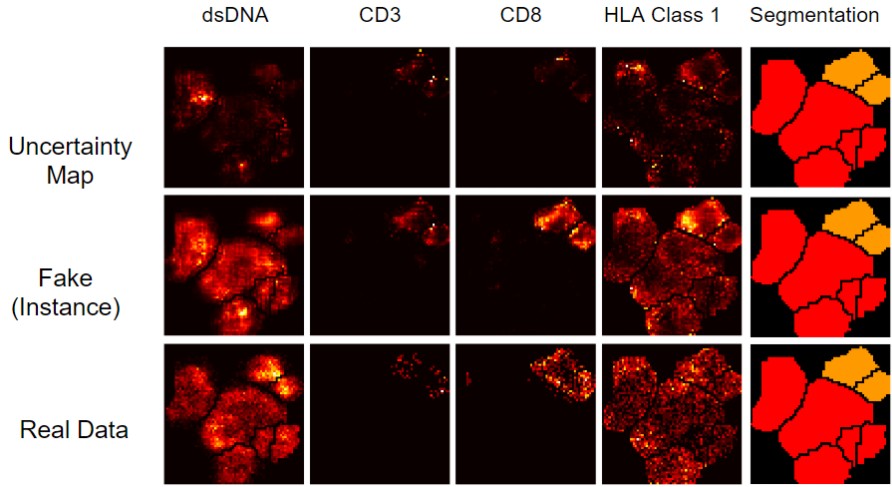

Figure 14: Uncertainty maps illustrating model generativeness.

## A.7 SEARCH ALGORITHM

Here we provide a randomized search algorithm to try to discover further cell-cell interactions in other channels.

| Index | Fixed Cell Type | Added Cell Type | Channels | Expression |
|---|---|---|---|---|
| 1 | CD8 T Cell | Keratin-positive tumor | PD-1, CD3, Vimentin | Increase ↑ |
| 2 | CD8 T Cell | CD8 T Cell | PD-1 | Decrease ↓ |
| 3 | CD3 T Cell | Keratin-positive tumor | CD3, CD4 | Increase ↑ |
| 4 | Keratin-positive tumor | CD8 T Cell | Pan-Keratin | Decrease ↓ |
| 5 | Keratin-positive tumor | CD3 T Cell | Pan-Keratin | Decrease ↓ |
| 6 | Keratin-positive tumor | Macrophages | Pan-Keratin | Decrease ↓ |
| 7 | Macrophages | Keratin-positive tumor | Vimentin | Increase ↑ |

Table 9: Additional cell interaction trends captured by CCIGAN.

The experimental findings in Indices 4, 5 and 6 support those reported in 6.2.2, demonstrating that immune cell presence adjacent to tumor cells causes a decrease in the tumor's pan-keratin, regardless of immune cell identity. In addition to confirming the results of 6.2.1, findings in indices 1 and 3 also indicate that tumor cells increase the expression of the T cell co-receptor, although this is of unclear functional significance. Index 2 suggests T cell clustering reduces the expression of the immune suppressive PD-1 marker. Lastly, Index 7 demonstrates an increase in macrophage expressed vimentin when macrophages are placed adjacent to tumor cells. Since vimentin is secreted as a pro-inflammatory marker in macrophages, this suggests an early macrophage inflammatory response to its tumor neighbor.

## A.8 MARKERS

The markers we used (total 24) in our experiments are: Pan-Keratin, EGFR, Beta catenin, dsDNA, Ki67, CD3, CD8, CD4, FoxP3, MPO, HLA-DR, HLA-Class-1, CD209, CD11b, CD11c, CD68, CD63, Lag3, PD1, PD-L1, IDO, Vimentin, SMA, CD31.

The markers we didn't use (total 12) in our experiments are: CD16, B7H3, CD45, CD45RO, Keratin17, CD20, CD163, CD56, Keratin6, CSF-1R, p53, CD138

**Algorithm 1:** Search Algorithm

**Input:** Cell segmentation list $\{S_i\}_{i=1}^n$, the channel index $m$, rotation angle $\Delta\theta$, fixed noise $\delta$, threshold $\beta$ and the generator $G$.

Randomly chose initial cell index $i_0 \in \{1, ..., n\}$;

Input segmentation $S^{\text{INPUT}} = S_{i_0}$;

Mask for the initial cell $U = \sum_{j=1}^n S_{i_0}[j, :, :]$ ;

Generated image $X_0 = G(\delta, S^{\text{INPUT}})$;

Specified channel $M_0 = U * X_0[m]$;

Show $S^{\text{INPUT}}$ and $M_0$;

**for** $k = 1$ **to** $n-1$ **do**

    Random index $i_k \in \{1, ..., n\}/ \cup_{j=0}^{k-1} \{i_j\}$ ;

    $S^{\text{INPUT}} = S^{\text{INPUT}} + S_{i_k}$ ;

    $X_k = G(\delta, S^{\text{INPUT}})$;

    $M_k = U * X_k[m]$;

    Show $S^{\text{INPUT}}$ and $M_k$;

    $E_k = d_{EM}(M_{k-1}, M_k)$;

    **if** $E_k > \beta \sum_{i,j} U_{i,j}$ **then**

        Log Significance;

        **if** $\sum_{i,j} M_{k-1,i,j} < \sum_{i,j} M_{k,i,j}$ **then**

            Log Increase;

        **else**

            Log Decrease;

