# OpenReview forum: "Exploring Cellular Protein Localization Through Semantic Image Synthesis"
_ICLR.cc/2020/Conference — Reject_

### Official Review · AnonReviewer2 · 2019-10-23
**Official Blind Review #2**

**Rating:** 3

**Review:**

The manuscript proposed a new method to model the data generated by multiplexed ion beam imaging by time-of-flight (MIBI-TOF). Essentially, the model leans the many-to-many mapping between the cell types and different protein markers' expression levels. Compared with the other mainstream GAN methods, the authors show the proposed method, CCIGAN, can outperform them in terms of generating the expression map of different protein markers given the segmentation of cell types. The manuscript also has an in-depth discussion of the biological meaning of the learned model as well as the learned vectors.

I personally like this manuscript a lot, considering the novelty and the thoroughness of the manuscript. However, I have the following concerns:

Major concern (The score will be significantly improved if the authors can handle these two concerns during revision):
1. My largest concern is how useful the proposed method is. It seems the wet experiments are mature. How can the computational synthesized image help the biologists then? Are there any cases that only the CCIGAN can do while the real experiment can not do? I am not an expert in MIBI-TOF, but I guess all the results in Section 6.2 can be easily achieved with the real data, right?
2. CCIGAN is indeed better than the other methods, but would it be accepted by the biologists in terms of performance? Would they believe in the results? In fact, in the last column of Figure 2, we can see clear artifacts for the results generated by CCIGAN.


Minor concern:
1. What's the resolution of the MIBI-TOF and CCIGAN?
2. The introduction of the background could be further refined. The many-to-many mapping between different cell types and different protein markers is not emphasized explicitly. Readers can get lost easily.
3. The data part is not clear. How many M*2048*2048 images do the authors have for training as well for testing?
4. Why do the authors choose 64*64 as the image path size?
5. Can the model be generalized well for data collected in different experiments (i.e., different tissues) but from the same machine? Can the model be generalized well across different machines with the same imaging experimental setting?
6. Is the model sensitive to the preprocessing of the data, like normalization? As for as I know, the baseline expression level of tissue can vary significantly at different time points within one day. If the model is sensitive to that, it will affect the usage of the model.

**Experience Assessment:**

I have read many papers in this area.

**Review Assessment: Checking Correctness Of Derivations And Theory:**

I assessed the sensibility of the derivations and theory.

**Review Assessment: Checking Correctness Of Experiments:**

I carefully checked the experiments.

**Review Assessment: Thoroughness In Paper Reading:**

I read the paper at least twice and used my best judgement in assessing the paper.

---

> ### Author Response · Authors · 2019-11-12
> **Response to Reviewer #2 Part 2**
>
>
> Minor concern:
> 1. What's the resolution of the MIBI-TOF and CCIGAN?
> MIBI TOF is 800 $mm^2$ at 2048x2048 pixels, doing some rearranging we see that 64x64 $\rightarrow$ 64/2048 *800 =  $25 mm^2$
>
> 2. The introduction of the background could be further refined. The many-to-many mapping between different cell types and different protein markers is not emphasized explicitly. Readers can get lost easily.
> Fixed! We also included Figures 1(B) (https://imgur.com/a/pKzch1J ), Figure 5 (https://imgur.com/a/1gckfA3 ) showing the purpose of the model of being able to pose arbitrary cell orientation scenarios. Additionally we have revised our evaluation portion to make it easier to understand what biological phenomenon we are corroborating and which metrics we used to do so.
>
> 3. The data part is not clear. How many M*2048*2048 images do the authors have for training as well for testing?
> 1 image with a 90-10 split on the image space to ensure no cell overlap. Now we have trained and included additional patients (each patient has one (M,2048,2048) image).
>
> 4. Why do the authors choose 64*64 as the image path size?
> This is a common output size, we can scale our model larger but computationally for other datasets.  This provides on average 8 cells, with a max of 15 and this captures enough interactions. Later on as we explore more datasets, we can scale this resolution up.
>
> 5. Can the model be generalized well for data collected in different experiments (i.e., different tissues) but from the same machine? Can the model be generalized well across different machines with the same imaging experimental setting?
> Yes the model is agnostic to the technology (for example CODEX, Vectra) as long as semantic segmentations are given. We are evaluating more patients to show cross patient tumor (infiltrated/non infiltrated) micro-environments that is of value to biologists. This will allow different hypothesis testing environments for understanding interactions between patients.
>
> 6. Is the model sensitive to the preprocessing of the data, like normalization? As for as I know, the baseline expression level of tissue can vary significantly at different time points within one day. If the model is sensitive to that, it will affect the usage of the model.
> Normalization matters in these experiments (current protocols stain together to limit batch effects between FOV, but this is experimental wet lab side). Yes normalization can affect samples and in turn the model, but this is all on the experimental side.  We are assuming these steps are done properly before.  Additionally, we are measuring  relatively within the patients. There are experimental techniques to try to mitigate these effects, such as patient samples being made into a TMA (tissue microarray)  in one slide and stained at the same time to try to minimize the batch effects. Autofluorescence will add some base noise, but MIBI does not do this, so signals are much less affected in terms of expression and non expression.
>
>
> Citations
> [1] Yichen Wu, Yair Rivenson, Hongda Wang, Yilin Luo, Eyal Ben-David, Laurent A Bentolila, Chris-tian Pritz, and Aydogan Ozcan. Three-dimensional virtual refocusing of fluorescence microscopy images using deep learning.Nature Methods, 2019.
> [2] Yoshiko Iwai, Masayoshi Ishida, Yoshimasa Tanaka, Taku Okazaki, Tasuku Honjo, and Nagahiro Minato. Involvement of pd-l1 on tumor cells in the escape from host immune system and tumor immunotherapy by pd-l1 blockade. PNAS, 19:12293–12297, 2002.
> [3] RG Oshima. Apoptosis and keratin intermediate filaments. Cell Death and Differentiation, 9:486– 492, 2002.
> [4] Keren, Leeat, et al. "A structured tumor-immune microenvironment in triple negative breast cancer revealed by multiplexed ion beam imaging." Cell 174.6 (2018): 1373-1387.

---

> ### Author Response · Authors · 2019-11-12
> **Response to Reviewer #2 Part 1**
>
> We thank the reviewer for their time and comments and would like to clarify and include the following in response to the listed major concerns:
>
> We’re glad you liked our manuscript! All of the major concerns (and additional experiments) are also reflected in the updated copy of our manuscript.
>
> 1. “Wet experiments are mature”
> Though wet lab experiments have significantly progressed, there are still major challenges in acquiring the data and interpreting it. Obtaining data is a time and labor intensive process and for patient data, we usually do not necessarily have nice controls while trying to interpret complicated interactions.  Furthermore there’s a trade off on resolution and quality for the data obtained; this is where we believe CCIGAN is useful. We have included an additional paragraph at the end of the introduction which clarifies the value of CCIGAN as an extension of MIBI-TOF data collection and its benefits for high throughput assessment of biological scenarios that would not be possible with wet experiments alone.
>
> “How can the computational synthesized image help the biologists then?  Are there any cases that only the CCIGAN can do while the real experiment can not do?”
> It seems that a possible source of confusion is the lack of clarity in the introduction. We have updated it appropriately. We have added a Figure 1(B) (https://imgur.com/a/pKzch1J ) to show the purpose of the model is that it allows us to pose counterfactual scenarios such as “what effect does adding cell type X next to cell type Y have on these two cells”.  This allows us to define a hypothesis testing environment for biologists without necessarily needing to scan multiple tissue slices to search for the exact scenarios. Theoretically given unlimited data, we can search for each cell-cell interaction scenario we wish to test, however, given the heterogeneous nature of cell interactions within tumor environments this is not a feasible approach. CCIGAN circumvents this hurdle posed by the true biological data in that it allows for us to construct any number of exacting cellular scenarios and output biologically consistent predictions. Because of this, the situations we have tested and quantified (e.g. varying number of cells, cell type, surface area through direct manipulation) cannot be easily (or feasibly) achieved by examining the real data. Recent work by Wu et al [1] have demonstrated the value of hypothesis testing in deep learning models in contributing to understanding of complex biological interactions. We have included this citation in our updated introduction as a demonstration of the additional value that deep learning models bring to the investigation of basic biological relationships.
>
> “all the results in Section 6.2 can be easily achieved with the real data, right?”
> No, this is not the case. Similarly to above, to find the situations we have tested (if it exists at all in the limited data), we would not have enough permutations and perturbations of the experiment to make a claim or to functionally quantify the experiment.
>
> 2. “would it be accepted by the biologists in terms of performance? Would they believe in the results?”
> While it is already difficult for anyone to trust a GAN without proper evaluation, by recapitulating prior accepted and clinically proven biological phenomenon under various control and test studies, we can demonstrate that CCIGAN learns biologically consistent insights and suggests that posing counterfactual scenarios are in turn, accurate. In addition to the current evaluation of recapitulating and newly quantifying previous established biological phenomenon (PD1/PD-L1 [2], CD8/pan keratin [3], section 5.2, 5.3), we have added another experiment to demonstrate that CCIGAN is able to capture and further quantify the same effects reported in MIBI-TOF’s triple negative breast cancer (TNBC) [4] study regarding tumor infiltrated and non infiltrated environments (section 5.4). This shows CCIGAN is able to recapitulate prior discovered biology from non-GAN methods and show the first steps towards quantifying them.
>
> “in the last column of Figure 2, we can see clear artifacts for the results generated by CCIGAN.”
> Good eye! While this may seem like an artifact, it’s actually a biologically consistent scenario of tumor expression of PD-L1. Any of the tumor cells (red) could potentially express PD-L1 as shown in the real data. While other models may learn this association, they do not learn it in a biologically consistent manner as demonstrated in section 5.3’s CD8/pan keratin experiment.

---

### Official Review · AnonReviewer1 · 2019-10-23
**Official Blind Review #1**

**Rating:** 3

**Review:**

This paper applies the SPADE semantic image synthesis technique (with a custom attention mechanism) to MIBI-TOF data to examine hypotheses about cell-to-cell interactions in the context of an immune infiltrated tumor sample. I think that this is potentially an interesting application of GANs -- to generalize beyond specific gathered data and instead start engaging with counterfactual scenarios like "what effect does it have to add cell type X next to cell type Y".

Unfortunately, I think this paper is not yet finished with regard to both (1) conveying sufficient motivation for the use of image synthesis for distilling biological insights and (2) evaluating the success or failure of the technique primarily in terms of generating useful biological insights. Simultaneously, there are enough red flags with regard to knowledge of the data generation and underlying biology that it's unclear if the authors could correctly sanity check any insights they extract from their model.

More specific criticisms & suggestions

1) It would help to provide much clearer motivation for applying image synthesis to MIBI-TOF data. Section 1.2 skips straight from describing the MIBI-TOF instrument to “we made a new kind of GAN”.  Again in the beginning of the Related Work the paper states: "We are interested in the task of generating biologically consistent expression patterns of cellular proteins given a segmentation map of cell neighborhoods”. But, why are synthetic images interesting given that we can actually look at real MIBI-TOF data. I start getting a sense of why this model might be interesting or useful only in Section 5 — more rationale is required earlier in the paper.

2) The evaluation section is hard to follow. I think it would be helpful to more clearly describe a larger set of biological phenomena that a practitioner would expect to see in the data, choose a single metric for each case, and show that that these phenomena are recapitulated. No one is going to trust a GAN to give them scientific insights unless they're very confident that all known / simple cell-to-cell interactions have a clear signal.

3) "MIBI-TOF bombards a tissue sample with elemental metals tethered to respective antibodies for dozens of distinct cellular proteins and detects each to obtain image channels” — this is not an accurate description of MIBI-TOF, at least not the instrument I'm familiar with. Typically the tissue is first stained with antibodies tagged with heavy metals and the instrument then bombards the tissue sample with simple ions (like O2+), causing the release of metal ions. It seems very unlikely that bombarding anything with antibody/metal conjugates could be informative.

4) "Tumor cells could be identified by markers such as pan-keratin and beta-catenin” — I guess in the context of a tumor sample beta-catenin could be over-expressed but it's present in pretty much every cell type, including lymphocytes (https://www.proteinatlas.org/ENSG00000168036-CTNNB1)

Small nits:

* Repeated use of “antibodies” in "Engineered antibodies for PD-1/PD-L1 antibodies” — maybe better to write “Antibodies which block the interaction of PD-1/PD-L1"

* "While MIBI-TOF is capable of 36 different markers, we discarded uninformative and irrelevant markers1 resulting in M = 24.” — I’m really surprised that someone put 12/36 uninformative markers in a MIBI-TOF panel. Aren’t these tagged antibodies expensive? Can we at least get a list of what got discarded?

* The model interpretability control seems weak in that lymphocytes are more likely to express MHC-I than tumor cells (which have a potential survival advantage from not expressing it) and tumor cells may actually have more dsDNA than lymphocytes due to changes in ploidy (or original differences in ploidy from e.g. liver cells).

**Experience Assessment:**

I have read many papers in this area.

**Review Assessment: Checking Correctness Of Derivations And Theory:**

I did not assess the derivations or theory.

**Review Assessment: Checking Correctness Of Experiments:**

I assessed the sensibility of the experiments.

**Review Assessment: Thoroughness In Paper Reading:**

I read the paper at least twice and used my best judgement in assessing the paper.

---

> ### Author Response · Authors · 2019-11-12
> **Response to Reviewer #1 Part 2**
>
>
> 3. Thanks for catching this! It was a miswriting error. We see how it can be construed as  MIBI-TOF bombards ((a tissue sample )(with elemental metals tethered to respective antibodies)) as opposed to (a tissue sample, with elemental metals tethered to respective antibodies). We have updated it to: Given a tissue sample that is first stained with antibodies tethered with elemental metals,  MIBI-TOF bombards the sample with simple ions causing the release of metal ions.
>
> 4. We have updated this to be less vague to “such as pan-keratin and the overexpression of beta-catenin”. In the MIBI-TOF TNBC paper, beta-catenin was used to classify tumor cells (Figure 1A,  [3]). We have also included the full list of markers to be more clear.
>
> Minor comments/Small nits:
>
> 1. We have updated some of the writing from the repeated use of “antibodies”
>
> 2. Thanks for pointing out our unclear writing. All the markers were used for cell typing in the original paper (where we obtained our classifications from). For our specific purposes, some of them are not interesting to us (i.e. CD45 is a status indicator and is too broad to study its localizations). They were useful for cell typing but not for studying cellular protein localization. We have included a list of markers in the appendix and provide a brief explanation on the unused markers (usually they would be empty or indicators for status). Even still, some of the markers we had selected, still ended up being blank/empty.
>
> Used (24): Pan-Keratin, EGFR, Beta catenin, dsDNA, Ki67, CD3, CD8, CD4, FoxP3, MPO, HLA-DR, HLA_Class_1, CD209, CD11b, CD11c, CD68, CD63, Lag3, PD1, PD-L1, IDO, Vimentin, SMA, CD31
> Not used (12): CD16, B7H3, CD45, CD45RO, Keratin17, CD20, CD163, CD56, Keratin6, CSF-1R, p53, CD138
>
> 3. Thanks for pointing this out! We see on the biological perspective, the typical TNBC patient has a single digit number of amplified segments which does not alter the overall amount of dsDNA than a few percent [7].
>
> For this portion we were trying to explain the learned vectors from a technical attention perspective. While this is biologically the case, we can empirically observe the average values to be very similar across HLA Class 1 (MHC-I) and dsDNA and similarly expressed across all cells (average MHC-I/HLA Class 1: 0.049, average dsDNA: 0.045). The purpose of the control was to show that switching the learned vectors in the attention (which are fitted to all cells) did not yield a difference as their attention was learned to be very similar as MHC-I, dsDNA are similarly expressed across all cells (contrasted with a switch between CD 8 and pan keratin vectors).
>
>
> Citations
> [1] Yoshiko Iwai, Masayoshi Ishida, Yoshimasa Tanaka, Taku Okazaki, Tasuku Honjo, and Nagahiro Minato. Involvement of pd-l1 on tumor cells in the escape from host immune system and tumor immunotherapy by pd-l1 blockade. PNAS, 19:12293–12297, 2002.
> [2] RG Oshima. Apoptosis and keratin intermediate filaments. Cell Death and Differentiation, 9:486– 492, 2002.
> [3] Keren, Leeat, et al. "A structured tumor-immune microenvironment in triple negative breast cancer revealed by multiplexed ion beam imaging." Cell 174.6 (2018): 1373-1387.
> [4] Weiping Zou. “Immunosuppressive networks in the tumour microenvironment and their therapeutic relevance.” Nature Reviews Cancer,  5, 263–274, 2005.
> [5] M Egelblad, E Nakasone, Z Werb. “Tumors as Organs: Complex Tissues that Interface with the Entire Organism.” Developmental Cell, 18(6), 884-901, 2010.
> [6] Hendrik Ungefroren, Susanne Sebens, Daniel Seidl, Hendrik Lehnert & Ralf Hass. “Interaction of tumor cells with the microenvironment.” Cell Communication and Signalling, 9(18), 2011.
> [7] Cancer Genome Atlas Network. "Comprehensive molecular portraits of human breast tumours." Nature 490.7418 (2012): 61.

---

> ### Author Response · Authors · 2019-11-12
> **Response to Reviewer #1 Part 1**
>
> We thank the reviewer for their time and thorough comments!
>
> (1) Not yet finished w.r.t. conveying sufficient motivation for the use of image synthesis for distilling biological insights:
>
> Similar to the later specific criticisms and suggestions 1), this stems from our previous introduction not being clear. We have revised our introduction to showcase the model’s ability to  “[engage] with counterfactual scenarios like ‘what effect does it have to add cell type X next to cell type Y’ ” (Figure 1(B), https://imgur.com/a/pKzch1J ). The motivation for image synthesis is that it allows us to distill more information (in a many to many mapping) from beyond just the cell type (segmentation map). As MIBI measures much more information in the protein channels (ie various localizations at a subcellular resolution), we want to see how predictive the neighborhoods and cell types are of a cell’s phenotype. Additionally through SPADE and our attention mechanism as an image synthesis technique, we are able to condition on surrounding cell neighbors and capture cell-cell interactions in a local receptive field. The role of CCIGAN as a step forward from MIBI-TOF data collection is more clearly stated earlier in the paper. Furthermore, CCIGAN’s value as a tool towards expediting high throughput assessments of cell-cell interactions is clarified, particularly with regard to the rationale of our research aim. This is reflected in our new introduction.
>
> (2) Not yet finished w.r.t. evaluating the success or failure of the technique primarily in terms of generating useful biological insights + there are enough red flags with regard to knowledge of the data generation and underlying biology that it's unclear if the authors could correctly sanity check any insights they extract from their model
>
> In addition to the current evaluation of recapitulating and newly quantifying previous established biological phenomenon (PD1/PD-L1 [1], CD8/pan keratin [2]), we have added another experiment to demonstrate that CCIGAN is able to capture and quantify the same effects reported in a study of triple negative breast cancer (TNBC) patients imaged by MIBI-TOF [3] publication regarding tumor infiltrated and non infiltrated environments (more on this later). This illustrates CCIGAN is able to recapitulate prior discovered biology from non-GAN methods and the first steps to quantify them.
>
> For data generation, the MIBI machine is limited because of the long time it takes to generate data. This is where CCIGAN is useful in creating counterfactual scenarios as opposed to searching for these instances (if any) in the real data. Additionally for CCIGAN, it is helpful in signal representation as it is mass-spec as opposed to fluorescence based where the latter has signals that are amplified and saturated. This leads to non quantitative measurements. Whereas in MIBI, each count is an individual protein. Because of the counts CCIGAN is able to learn a more biologically consistent representation.
>
> Sanity checking through quantifying and evaluating images is a biologically difficult problem at a cellular level, and a more difficult problem at a subcellular level. In addition to our previous our evaluation process of biologically motivated metrics where we spatially quantify direction and orientation through weighted centroid vector analysis and Earth Mover’s Distance, we use a simpler expression ratio sum to demonstrate we capture the same biological phenomena as [3] in the additional added (non) infiltration experiment.
>
> For specific criticisms and suggestions:
>
> 1. Thanks for pointing this out! We have revised our introduction, related works, and added Figure 1(B) exhibiting specific counterfactual scenarios that can be posed to CCIGAN.
>
> 2. We have moved the bulk of the evaluation section to the appendix and have restructured section 5 and 6 to provide the biological phenomena and a high level overview of the metric chosen to evaluate it. We hope that this makes it easier to follow. We wholeheartedly agree that trusting a GAN is a bad idea without proper evaluation. In addition to the previous PD1/PDL1 [1], CD8/pan keratin [2] experiments we have updated the results portion (point (2)) to incorporate a MIBI-TOF TNBC study [3].
>
> Unfortunately a global bank illustrating simple cell-cell interactions does not exist and even then, simple cell-cell interactions within the tumor microenvironment are quite complicated in that they have multiple higher order causal factors [4-6]. Furthermore, many of the markers in the original MIBI cohort were used for cell identification, not studying protein localization. The best a model can do to demonstrate biological consistency and establish trust (within the scope of ICLR) is to recapitulate known, vetted, and accepted biological phenomena. Through our now 3 independent experiments we believe we have demonstrated biological consistency and lay the foundation for interpreting cell-cell interactions through GAN methods.

---

### Official Review · AnonReviewer3 · 2019-10-28
**Official Blind Review #3**

**Rating:** 6

**Review:**

The authors present a GAN for multiplexed imaging (MIBI-TOF) data called CCIGAN. They propose an interesting architecture design with protein-specific attention to find association between cell types and neighboring pattens and cell-cell interactions. They also propose new and biologically interpretable metrics including a reconstruction metric, projected EMD and regressing of expression on neighbors.

They present improved reconstruction of interactions compared to other models in the context of PD-1 and PD-L1 interactions. It would be great to extend the evaluation to other interactions and tissue types.

Overall the paper is well written, the application and especially the focus on cell-cell interactions is novel. The model is properly justified and evaluated, and there is a high demand for this framework in the multiplexed imaging field.


**Experience Assessment:**

I have read many papers in this area.

**Review Assessment: Checking Correctness Of Derivations And Theory:**

I did not assess the derivations or theory.

**Review Assessment: Checking Correctness Of Experiments:**

I assessed the sensibility of the experiments.

**Review Assessment: Thoroughness In Paper Reading:**

I read the paper at least twice and used my best judgement in assessing the paper.

---

> ### Author Response · Authors · 2019-11-12
> **Response to Reviewer #3**
>
> We thank the reviewer for their time and comments and would like to include the following:
>
> 1. “It would be great to extend the evaluation to other interactions and tissue types.”
>
> Because MIBI only has 1 dataset available (as of November 2019), other tissues would be very difficult or impossible to compare/get. However, we can do further studies related to the different tumor scenarios across the patient dataset. We have added additional experiments in showing that we can recapitulate further established biological phenomena information beyond PD-1/PD-L1 [1], Pan-Keratin/CD8 [2] interactions (Section 5.2, 5.3).
>
> In particular we added an experiment to demonstrate we can further recapitulate a study done on TNBC data captured by MIBI-TOF [3] (Section 5.4). We added macrophage, tumor, and various T-cell interactions in tumor infiltrated environments and tumor compartmentalized environments for different patients (Figure 1 (B)). We find that CCIGAN is able to not only recapitulate the results in tumor infiltrated and non infiltrated environments discovered in the TNBC data [3], but also quantify them at a subcellular level.
>
> 2. “Overall the paper is well written, the application and especially the focus on cell-cell interactions is novel. The model is properly justified and evaluated, and there is a high demand for this framework in the multiplexed imaging field.”
>
> Thanks for the positive comments! We have further refined and restructured the introduction to make our contributions more apparent, such as CCIGAN’s purpose of offering a model for quick in silico hypothesis testing of counterfactual cell-cell interaction scenarios.  We have also moved the bulk of the evaluation criteria (section 5) into the appendix and combined Section 5 and 6 for easier reading to showcase more cell-cell interactions and their corresponding evaluation methods.
>
> Citations
> [1] Yoshiko Iwai, Masayoshi Ishida, Yoshimasa Tanaka, Taku Okazaki, Tasuku Honjo, and Nagahiro Minato. Involvement of pd-l1 on tumor cells in the escape from host immune system and tumor immunotherapy by pd-l1 blockade. PNAS, 19:12293–12297, 2002.
> [2] RG Oshima. Apoptosis and keratin intermediate filaments. Cell Death and Differentiation, 9:486– 492, 2002.
> [3] Keren, Leeat, et al. "A structured tumor-immune microenvironment in triple negative breast cancer revealed by multiplexed ion beam imaging." Cell 174.6 (2018): 1373-1387.

---

### Author Response · Authors · 2019-11-12
**Overview of changes**

We would like to thank all the reviewers for their time and comments/suggestions. As a high level overview, we added an experiment from a clinical MIBI-TOF study [1] and demonstrate that CCIGAN is able to further recapitulate and quantify the discovered biology (Section 5.4 in addition to Sections 5.2, 5.3). We also addressed the biological concerns and clarity with our manuscript. Additionally, we have rewritten the introduction and related works to further clarify our contributions and motivations for image synthesis. Lastly, we combined sections 5 (evaluation metrics) and 6 (biological significance) into a single section (section 5) for easier reading.

[1] Keren, Leeat, et al. "A structured tumor-immune microenvironment in triple negative breast cancer revealed by multiplexed ion beam imaging." Cell 174.6 (2018): 1373-1387.

---

### Decision · Program_Chairs · 2019-12-19

**Decision:**

Reject

**Comment:**

This paper proposes a dedicated deep models for analysis of multiplexed ion beam imaging by time-of-flight (MIBI-TOF).

The reviewers appreciated the contributions of the paper but not quite enough to make the cut.

Rejection is recommended.